# The allosteric modulation of complement C5 by knob domain peptides

Alex Macpherson[1,2]\*, Maisem Laabei[2], Zainab Ahdash[1], Melissa A Graewert[3], James R Birtley[1], Monika-Sarah ED Schulze[1], Susan Crennell[2], Sarah A Robinson[4], Ben Holmes[1], Vladas Oleinikovas[1], Per H Nilsson[1,5,6], James Snowden[1], Victoria Ellis[1], Tom Eirik Mollnes[6,7,8], Charlotte M Deane[4], Dmitri Svergun[3], Alastair DG Lawson[1], Jean MH van den Elsen[2,9]\*

[1]UCB, Slough, United Kingdom; [2]Department of Biology and Biochemistry, University of Bath, Bath, United Kingdom; [3]European Molecular Biology Laboratory, Hamburg Unit, Hamburg, Germany; [4]Department of Statistics, University of Oxford, Oxford, United Kingdom; [5]Department of Chemistry and Biomedicine, Linnaeus University, Kalmar, Sweden; [6]Department of Immunology, Oslo University Hospital, University of Oslo, Oslo, Norway; [7]Research Laboratory, Bodø Hospital, K.G. Jebsen TREC, University of Tromsø, Tromsø, Norway; [8]Centre of Molecular Inflammation Research, Norwegian University of Science and Technology, Trondheim, Norway; [9]Centre for Therapeutic Innovation, University of Bath, Bath, United Kingdom

**Abstract** Bovines have evolved a subset of antibodies with ultra-long heavy chain complementarity determining regions that harbour cysteine-rich knob domains. To produce high-affinity peptides, we previously isolated autonomous 3–6 kDa knob domains from bovine antibodies. Here, we show that binding of four knob domain peptides elicits a range of effects on the clinically validated drug target complement C5. Allosteric mechanisms predominated, with one peptide selectively inhibiting C5 cleavage by the alternative pathway C5 convertase, revealing a targetable mechanistic difference between the classical and alternative pathway C5 convertases. Taking a hybrid biophysical approach, we present C5-knob domain co-crystal structures and, by solution methods, observed allosteric effects propagating >50 Å from the binding sites. This study expands the therapeutic scope of C5, presents new inhibitors, and introduces knob domains as new, low molecular weight antibody fragments, with therapeutic potential.

\*For correspondence:
Alex.MacPherson@ucb.com (AM);
bssjmhve@bath.ac.uk (JMHE)

## Introduction

By the end of 2019, over 60 peptide drugs have received regulatory approval, with an estimated 400 more in active development globally (*Lau and Dunn, 2018*; *Lee et al., 2019*). As a potential route to discover therapeutic peptides, we previously reported a method for deriving peptides from the ultra-long heavy chain complementarity determining region 3 (ul-CDRH3), which are unique to a subset of bovine antibodies (*Macpherson et al., 2020*). We have shown that knob domains, a cyste-ine-rich mini-domain common to all ul-CDRH3, can bind antigen autonomously when removed from the antibody scaffold (*Macpherson et al., 2020*). This allows peptide affinity maturation to be performed in vivo, harnessing the cow's immune system to produce peptides with complex stabilising networks of disulphide bonds.

For the discovery of knob domain peptides, immunisation of cattle is followed by cell sorting of B-cells using fluorescently labelled antigen. A library of antigen-specific CDRH3 sequences is created by performing a reverse transcription polymerase chain reaction (RT PCR) on the B-cell lysate,

**eLife digest** Antibodies are proteins produced by the immune system that can selectively bind to other molecules and modify their behaviour. Cows are highly equipped at fighting-off disease-causing microbes due to the unique shape of some of their antibodies. Unlike other jawed vertebrates, cows' antibodies contain an ultra-long loop region that contains a 'knob domain' which sticks out from the rest of the antibody. Recent research has shown that when detached, the knob domain behaves like an antibody fragment, and can independently bind to a range of different proteins.

Antibody fragments are commonly developed in the laboratory to target proteins associated with certain diseases, such as arthritis and cancer. But it was unclear whether the knob domains from cows' antibodies could also have therapeutic potential. To investigate this, Macpherson et al. studied how knob domains attach to complement C5, a protein in the inflammatory pathway which is a drug target for various diseases, including severe COVID-19.

The experiments identified various knob domains that bind to complement C5 and inhibits its activity by altering its structure or movement. Further tests studying the structure of these interactions, led to the discovery of a common mechanism by which inhibitors can modify the behaviour of this inflammatory protein.

Complement C5 is involved in numerous molecular pathways in the immune system, which means many of the drugs developed to inhibit its activity can also leave patients vulnerable to infection. However, one of the knob domains identified by Macpherson et al. was found to reduce the activity of complement C5 in some pathways, whilst leaving other pathways intact. This could potentially reduce the risk of bacterial infections which sometimes arise following treatment with these types of inhibitors.

These findings highlight a new approach for developing drug inhibitors for complement C5. Furthermore, the ability of knob domains to bind to multiple sites of complement C5 suggests that this fragment could be used to target proteins associated with other diseases.

followed by a PCR using primers specific to the conserved framework regions which flank CDRH3 (*Macpherson et al., 2020*). Upon sequencing, ul-CDRH3s are immediately evident and the knob domains can be expressed recombinantly as cleavable fusion proteins (*Macpherson et al., 2020*).

This method for the discovery of knob domain peptides was established using complement component C5, and we reported peptides which bound C5 with affinities in the pM–low nM range (*Macpherson et al., 2020*). Herein, we use these novel peptides to probe the structural and functional aspects of C5 activation.

C5 is the éminence grise of the complement cascade's druggable proteins, and the target of effective therapies for diseases with pathogenic complement dysregulation, of which paroxysmal nocturnal haemoglobinuria (*Rother et al., 2007*) and atypical haemolytic uraemic syndrome (*Nürnberger et al., 2009*) are notable examples. Six monoclonal antibodies targeting C5 have reached, or are entering, clinical trials, closely followed by C5-targeting immune evasion molecules (*Romay-Penabad et al., 2014*), aptamers (*Biesecker et al., 1999*), cyclic peptides (*Ricardo et al., 2014*), interfering RNA (*Borodovsky et al., 2014*), and small molecules (*Jendza et al., 2019*). Currently, C5 inhibitors are being trialled for the treatment of acute respiratory distress syndrome arising from severe acute respiratory syndrome coronavirus 2 (SARS-CoV-2) infection (*Smith et al., 2020*; *Wilkinson et al., 2020*; *Zelek et al., 2020*) and for the neuro-muscular disease myasthenia gravis (*Albazli et al., 2020*).

C5 is the principal effector of the terminal portion of the complement cascade. At high local C3b concentrations, arising from activation of either or both of the classical (CP) and mannose binding lectin (LP) pathways, aided by the amplificatory alternative pathway (AP), C5 is cleaved into two moieties with distinct biological functions. Cleavage is performed by two convertases: C4bC2aC3b, formed in response to CP or LP activation (*Takata et al., 1987*) (henceforth the CP C5 convertase), and C3bBbC3b, formed in response to AP activation (*DiScipio, 1981*) (henceforth the AP C5 convertase). Although the constitutive components of the C5 convertases differ, they are thought to be

mechanistically identical. Once cleaved, the C5a fragment is the most proinflammatory anaphylatoxin derived from the complement cascade. When signalling through C5aR1 and C5aR2, C5a is a strong chemoattractant recruiting neutrophils, eosinophils, monocytes, and T lymphocytes to sites of complement activation, whereupon it activates phagocytic cells, prompting degranulation. C5b, meanwhile, interacts with C6, recruiting C7–C9 to form the terminal C5b-9 complement complex or TCC (*Lachmann and Thompson, 1970*). Once inserted into a cell membrane, the TCC is referred to as the membrane attack complex (MAC), a membrane-spanning pore which can lyse sensitive cells (*Götze and Müller-Eberhard, 1970*).

Aspects of the structural biology of C5 are well understood due to a crystal structure of the apo form (*Fredslund et al., 2008*) and a number of co-crystal structures of C5 with various modulators. By virtue of its constitutive role in the terminal pathway, C5 is a recurrent target for immune evasion molecules and structures have been determined of C5 in complex with an inhibitory molecule derived from *Staphylococcus aureus*, SSL-7 (*Laursen et al., 2010*), as well as several structurally distinct examples from ticks: OmCI (*Jore et al., 2016*), RaCI (*Jore et al., 2016*), and Cirp-T (*Reichhardt et al., 2020*). Additionally, the structures of C5 with the inhibitory monoclonal antibody (mAb) eculizumab (*Schatz-Jakobsen et al., 2016*), of C5 with a small molecule inhibitor (*Jendza et al., 2019*), and of C5 with the complement-depleting agent cobra venom factor (CVF) (*Laursen et al., 2011*) have all been determined.

Here, we probe C5 with knob domain peptides and explore the molecular processes which underpin allosteric modulation of this important drug target. This study is the first to investigate the molecular mechanisms and pharmacology of this recently isolated class of peptide.

## Results

### Bovine knob domain peptides as potential C5 inhibitors

We have previously shown that antigen-specific, disulphide-rich knob domain peptides derived from bovine antibodies have great potential for therapeutic utility. Using this approach, we obtained four knob domain peptides: K8, K57, K92, and K149, which we have shown to display tight binding to human C5. Previously we reported equilibrium dissociation constants of 17.8 nM for K8, 1.4 nM for K57, <0.6 nM for K92, and 15.5 nM for K149 (*Macpherson et al., 2020*).

### Functional characterisation of anti-C5 bovine knob domain peptides

For functional characterisation of the peptides, we performed complement assays for CP and AP activation in human serum, assessing C5b neo-epitope formation and C5a release (schematically presented in *Figure 1A, B*), in combination with orthogonal ELISAs, measuring C3b and C9 deposition. Here, we show that K57 was a potent and fully efficacious inhibitor of C5 activation, preventing release of C5a, and deposition of C5b and C9. As expected, there was no effect on C3b, which is upstream of C5 (*Figure 1C, D*). In contrast, K149 was a high-affinity silent binder with no discernible effect on C5a release, formation of C5b neo-epitope or C9 deposition, even at peptide concentrations in excess of $100 \times K_D$ (see *Supplementary file 1* Section 1).

K8 and K92 exerted more nuanced allosteric effects on C5 (*Figure 1C, D*). By ELISA, K92 partially prevented C5 activation by the AP, but, intriguingly, no effect was observed in CP assays, suggesting K92 selectively inhibits C5 activation by the AP C5 convertase, but not the CP C5 convertase. Partial antagonists, where the degree of inhibition for the asymptotic concentrations of a dose–response curve ($E_{max}$) is below 100%, are an impossible mode of pharmacology for orthosteric antagonists (*Klein et al., 2013*), and we therefore propose that K92 operates by a non-steric mechanism. K8 was also demonstrably allosteric, partially inhibiting both the AP and CP in ELISA experiments. For K8 and K92, no effect on C3b deposition was detected.

When tested in CP and AP haemolysis assays (*Figure 1E, F*), K57 was a potent and fully efficacious inhibitor of complement-mediated cell lysis. Consistent with the ELISA data, K92 was active solely in the AP-driven haemolysis assay, achieving $E_{max}$ values of 30–40%; while K8 was efficacious in the CP assay but did not show activity in the AP assay below 10 µM, potentially a consequence of the increased serum concentration and stringency of the haemolysis endpoint.

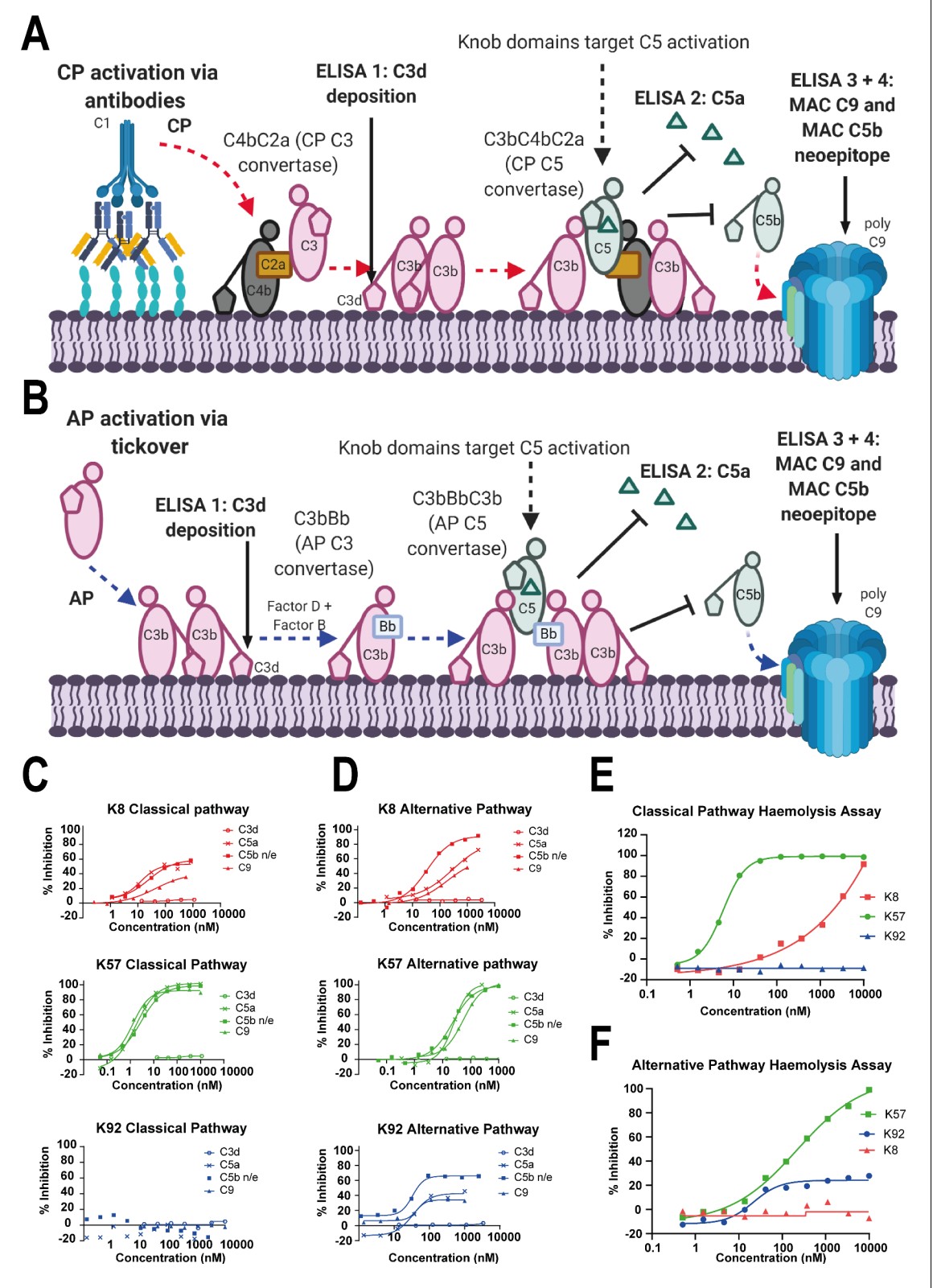

**Figure 1.** Functional modulation of C5 via knob domain peptides. (**A**) shows an abridged schematic for classical pathway (CP) activation. Following activation of C1q via antibody Fc, C4 and C2 are cleaved and form C4bC2a (the CP C3 convertase) which cleaves C3 into C3a (not shown) and C3b. At high C3b concentrations, C4bC2aC3b (the CP C3 convertase) forms and cleaves C5 into C5a and C5b. C5b associates with C6 and forms the membrane attack complex (MAC) with C7, C8, and multiple copies of C9. (**B**) shows an abridged schematic for surface phase alternative pathway (AP)

*Figure 1 continued on next page*

Figure 1 continued

activation in assays (where generation of C3b from the CP/LP is virtually excluded); tick-over of C3 generates C3a (not shown) and C3b. In the presence of factor B and factor D, C3bBb (the AP C3 convertase) generates additional C3b, prompting formation of C3bBbC3b (the AP C5 convertase), which cleaves C5 into C5a and C5b, driving MAC formation. CP-driven ELISAs (C) and AP-driven ELISAs (D) are shown. For both pathways, the inhibition of C3d (the surface-associated domain of C3b, which is upstream of C5 inhibition), C5a release, and C5b neo-epitope formation and C9 deposition were tracked within the MAC. Haemolysis assays with sheep erythrocytes, for the CP (E), and rabbit erythrocytes, for the AP (F), show that K57 is a potent and efficacious inhibitor of both pathways. K92 is selective, partial antagonist of the AP, while K8 is a weak antagonist of the CP but did not show efficacy in the AP haemolysis assay, below 10 μM. For the AP assays, 5% serum (v/v) gives a putative C5 concentration of 20 nM. For the CP assays, 1% serum (v/v) gives a putative C5 concentration of 4 nM, based on a reported C5 serum concentration of 397 nM/75 μg/mL (*Sjöholm, 1975*).

## Cooperativity in C5 binding by knob domain peptides

To test for cross blocking, arising from overlapping epitopes, or cooperativity between knob domains, we performed a surface plasmon resonance (SPR) cross blocking experiment, where, using a Biacore 8K, we saturated a C5-coated sensor chip with two 20 μM injections of knob domain peptide before injecting a different peptide at 20 μM to assess its capacity to bind. This provides a qualitative measure of cross blocking, whereby an increase in response units (RUs) indicates ternary complex formation, stoichiometries, or kinetics cannot be reliably derived with concurrent dissociation of both peptides (*Figure 2*).

Saturation of C5 with K8, K57, or K92 did not prevent subsequent binding of the non-functional K149 (*Figure 2D*), suggesting K149 does not share an epitope with the other ligands, nor does it significantly perturb C5 such that the other binding sites are affected.

We detected negative cooperativity between K8 and K92, whereby saturation of C5 with K8 entirely prevented binding of K92. When the order of addition was changed and C5 was saturated with K92, K8 was still able to bind, albeit to a lesser degree (*Figure 2B*). Saturation of C5 with K8 also entirely eliminated binding of K57, with a similar order of addition effect, whereby K8 could still partially bind to the C5-K57 complex (*Figure 2A*). When C5 was saturated with K92 or K57, only very small amounts of subsequent binding of either peptide were observed by SPR (*Figure 2C*), suggesting that the epitopes do not overlap but that considerable negative cooperativity exists.

## Structural analysis of C5-knob domain complexes

### Crystal structure of the C5-K8 peptide complex

To elucidate the structural basis for the allosteric modulation of C5, we determined the crystal structure of the C5-K8 complex at a resolution of 2.3 Å (see *Supplementary file 1,* Table 2.1 for data collection and structure refinement statistics). The structure of the C5-K8 complex shows the K8 peptide binding to a previously unrecognised regulatory site on C5; the macroglobulin (MG) 8 domain of the α-chain (*Figure 3A*). K8 adopts a cysteine knot-like configuration, where a flattened 3-strand β-sheet topology is constrained by three disulphide bonds (*Figure 3A* and *Figure 3—figure supplement 1A*). Analysis of the K8-C5 complex with the macromolecular interfaces analysis tool PDBePISA (*Krissinel and Henrick, 2007*) reveals a large interaction surface (total buried surface area in complex: 1642 Å$^2$; with 852 Å$^2$ contributed by K8 and 790 Å$^2$ by C5), comparable to those seen in Fab-antigen complexes (*Ramaraj et al., 2012*), stabilised by an extensive network of 18 hydrogen bonds between K8 and the MG8 domain (*Figure 3A* and *Supplementary file 1*), dominated by arginine residues R23$_{K8}$, R32$_{K8}$, and R45$_{K8}$. The extensive H-bond network is further bolstered by several ionic interactions, between R32$_{K8}$ and D1471$_{C5}$ (C5 numbering based on mature sequence), D25$_{K8}$ and K1409$_{C5}$, and H36$_{K8}$ and D1382$_{C5}$ (*Figure 3A* and *Supplementary file 1* Table 2.3). The opposing face of K8 was fortuitously stabilised by a substantial, 1275 Å$^2$, crystal contact with the C5d domain of a symmetry-related C5 molecule (*Figure 3—figure supplement 1B*), ensuring low relative B-factor values (K8: 58 Å$^2$, C5-K8 complex: 65 Å$^2$) (see also *Figure 3—figure supplement 1C*) and clear and continuous electron density, enunciating the unique disulphide bond arrangement of the knob domain peptide and the backbone and side chains interactions with C5 (*Figure 3—figure supplement 2A* shows a mFo-DFc simulated annealing OMIT map of the C5-K8 complex). Despite the overall resolution of the dataset comparing favourably with other C5 structures in the PDB (*Schatz-Jakobsen et al., 2016*; *Laursen et al., 2010*; *Jore et al., 2016*; *Fredslund et al., 2008*), density for the C345c domain was largely absent due to this flexible domain occupying a solvent channel. This flexible attachment of the C345c domain to the α-chain of C5 is

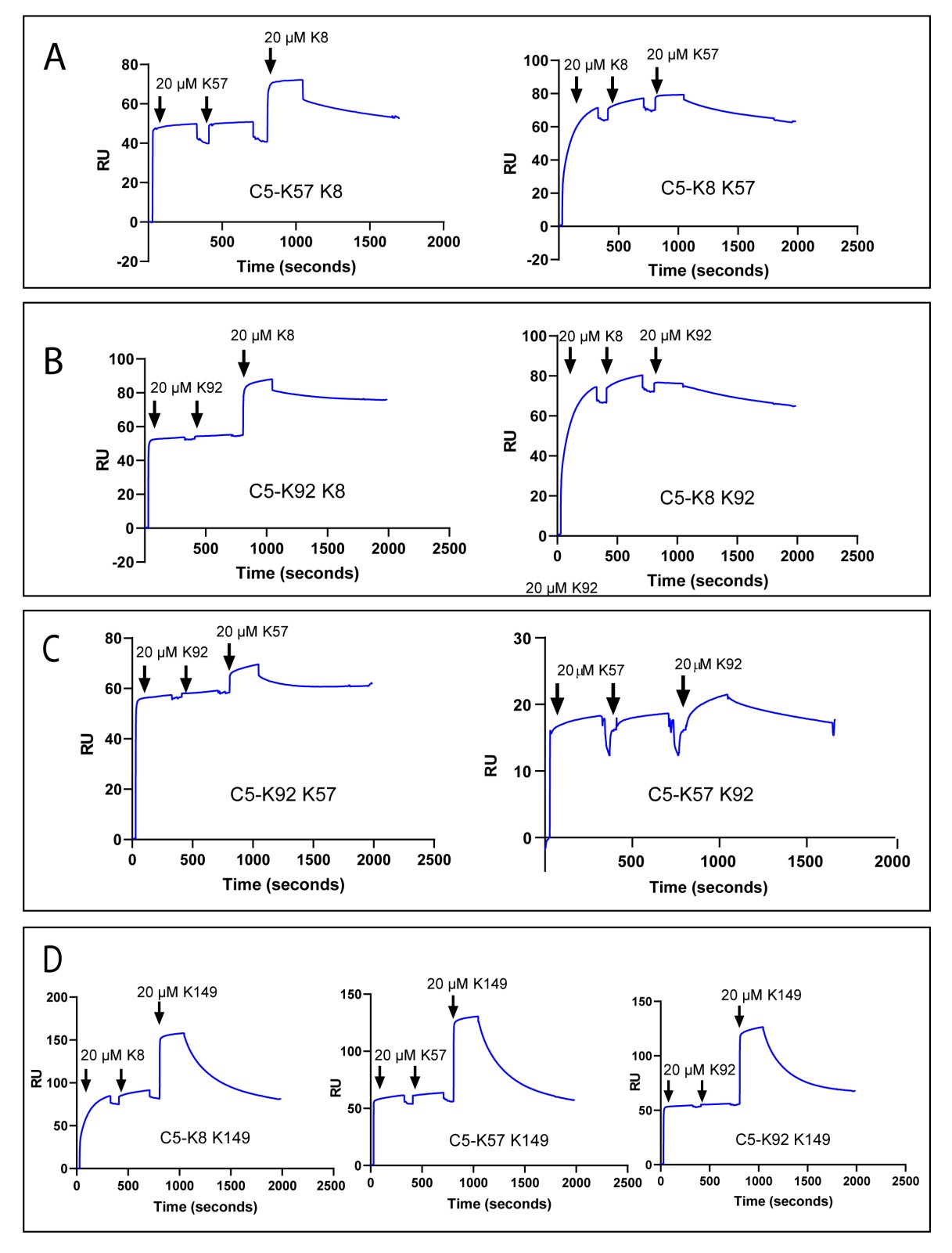

**Figure 2.** Surface plasmon resonance peptide cross blocking. (**A**), (**B**), and (**C**) highlight negative cooperativity between the K8, K92, and K57 peptides, respectively. Neither K57 or K92 can bind to the C5-K8 complex but K8 can bind, albeit at a lower level, to C5-K57 and C5-K92. We could not detect any negative cooperativity between K8, K57, or K92 with the silent binder K149, shown in (**D**). RU: response unit.

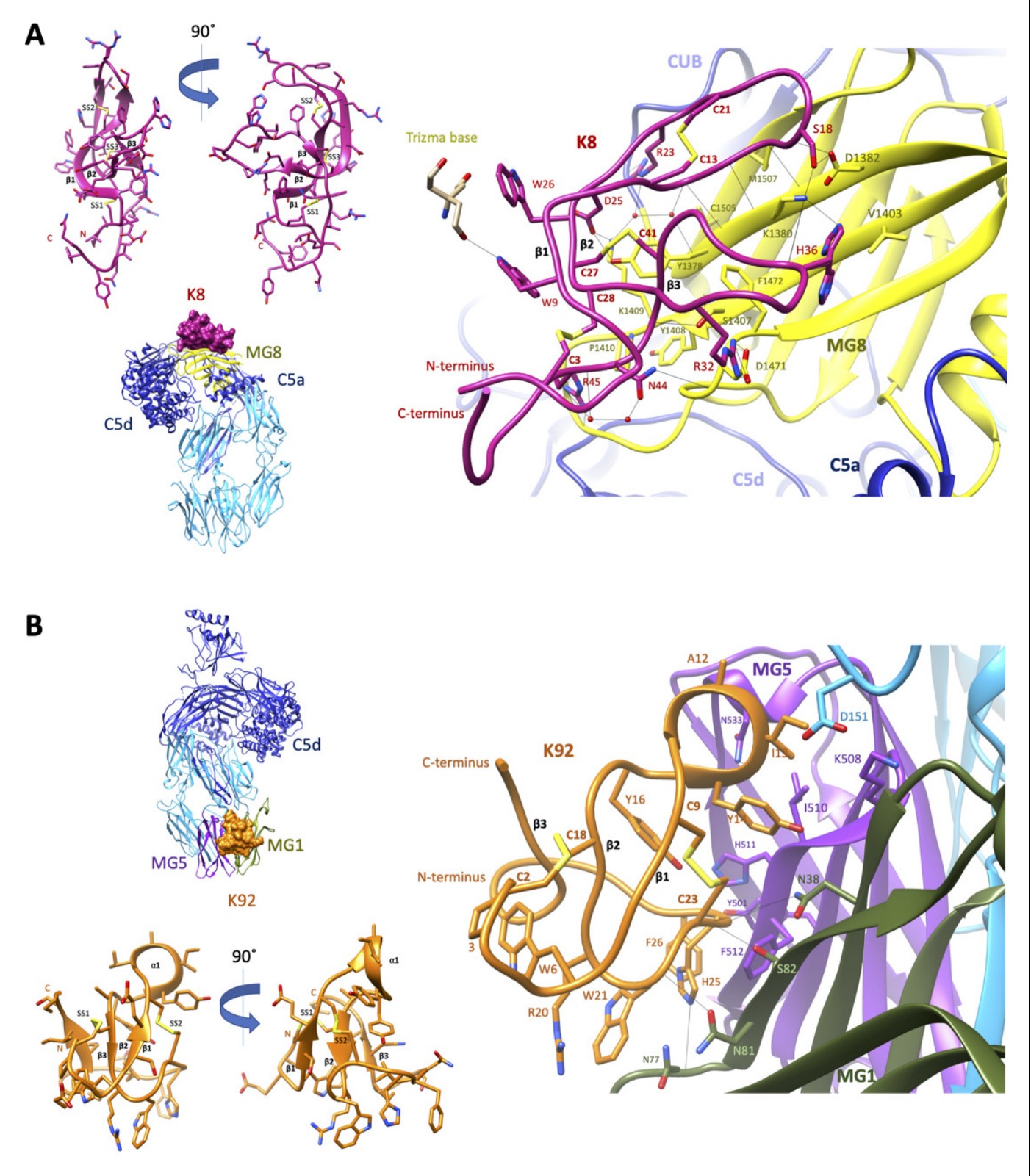

**Figure 3.** Crystal structures of C5-knob domain complexes. (**A**) and (**B**) show the crystal structures of C5 in complex with the K8 and K92 knob domain peptides, respectively. The binding site for the K8 peptide (**A**, shown in red) is located on a previously unreported ligand binding site on the macroglobulin (MG) 8 domain (shown in yellow) of C5. The binding site for K92 (**B**, shown in orange) is located between the MG1 and MG5 domains (shown in green and magenta, respectively).

*Figure 3 continued on next page*

eLife Research article

Immunology and Inflammation | Structural Biology and Molecular Biophysics

*Figure 3 continued*

The online version of this article includes the following figure supplement(s) for figure 3:

**Figure supplement 1.** Structural analysis of the C5-K8 complex.
**Figure supplement 2.** Simulated annealing OMIT maps of the C5-knob domain peptide complexes.
**Figure supplement 3.** Structural analysis of the C5-K92 complex.
**Figure supplement 4.** Paratope analysis of K8 and K92 peptides.

observed consistently across the C5 structures (*Schatz-Jakobsen et al., 2016*; *Laursen et al., 2010*; *Jore et al., 2016*; *Fredslund et al., 2008*).

## Crystal structure of the C5-K92 complex

We also present a crystal structure of the C5-K92 complex at a resolution of 2.75 Å (see *Supplementary file 1* for data collection and structure refinement statistics). Continuous electron density for the flexible C345c domain of C5 was observed due to it being stabilised in an upward pose by crystal contacts, akin to the C5-RaCI-OmCI ternary complex structures (Protein Data Bank [PDB] accession codes 5HCC, 5HCD, and 5HCE; *Jore et al., 2016*). Despite displaying higher affinity binding to C5 than K8, electron density for K92 was less well defined as it occupies a solvent channel, and stabilising crystal packing interactions are absent (*Figure 3—figure supplement 3B*). A mFo-DFc simulated annealing OMIT map of the C5-K92 complex is displayed in *Figure 3—figure supplement 2B*, showing clear but sparse electron density for the peptide at 1.3 σ. Correspondingly, only a small increase in $R_{free}$, of 25.35–25.53, is observed when the peptide is removed during refinement compared to an increase in $R_{free}$ of 23.36–27.08 upon removal of the K8 peptide, potentially indicating that the occupancy is significantly below 1. This is also reflected in the high relative B-factor values for K92 (182.5 $Å^2$) compared to that of the complex (100.5 $Å^2$) (see also *Figure 3—figure supplement 3C*). Model building of the K92 peptide was aided by disulphide mapping using mass spectrometry. The disulphide map of K92 identified formation of disulphide bonds between $C9_{K92}$ and $C23_{K92}$ and between $C2_{K92}$ and $C18_{K92}$ (*Supplementary file 1*, Table 2.4), enabling completion of the model.

Similar to K8, K92 adopts a 3-strand β-sheet topology (*Figure 3B* and *Figure 3—figure supplement 3A*) but stapled with only two disulphide bonds. With shorter β-strands and longer connecting loop regions, K92 exhibits a more compact, globular arrangement (*Figure 3B*). Two extended loop regions interact with C5, including an α-helix containing loop between β-strands 1 and 2, occupying a cleft between the MG1 and MG5 domains of the β-chain of C5. The interaction surface (total buried surface area: 1365 $Å^2$; with 750 $Å^2$ contributed by K92 and 615 $Å^2$ by C5) is sustained via a sparse set of eight H-bonds (*Supplementary file 1*, Table 2.5). A series of π–π and aliphatic–aromatic stacking interactions spans K92, encompassing $F26_{K92}$, $H25_{K92}$, $W21_{K92}$, $W6_{K92}$, and $P3_{K92}$ (*Figure 3B*). From within this hydrophobic patch, H-bonds occur between $H25_{K92}$ and the backbone carbonyls of $N77_{C5}$ and $N81_{C5}$ on the MG1 domain.

## Validation of the observed C5-peptide complexes

To validate the observed K8 and K92 C5-binding modes observed in our crystal structures, we assessed the binding properties for a number of alanine mutants of K8 and K92. For K8, R23A and R32A mutants targeted the two salt-bridge interactions with C5. While for K92, where there were few electrostatic interactions mediated by side chains, we targeted a hydrogen bond, sustained by H25, and important hydrophobic interactions with C5, involving neighbouring aromatics W21 and F26. While the K92 H25A mutant could not be expressed, the other mutants were tested, alongside unmodified K8 and K92, in SPR multi-cycle kinetics experiments (n = 3). For K8, the R23A resulted in modest twofold decrease in affinity, but R32A was markedly more attenuating, with a 715-fold drop in affinity (*Supplementary file 1,* Table 2.6). For K92, the loss of hydrophobic interactions with C5 in W21A and F26A mutants markedly abridged affinity with a 1209.2-fold and 45.7-fold drop in affinity, respectively (*Supplementary file 1,* Table 2.6).

To analyse the interfaces observed in the structures, we performed binding pose metadynamics (*Clark et al., 2016*), an analysis typically employed to computationally evaluate the binding stability of chemical ligands (*Fusani et al., 2020*). This in silico analysis suggested that both

K8 and K92's binding poses were exceptionally stable, with the interface maintaining the key interactions in spite of applied force (*Supplementary file 1*, Tables 2.7 and 2.8). This, in conjunction with earlier kinetic studies (*Macpherson et al., 2020*), highlights the stability of the interactions made by both knob domains.

### Cysteines participate in inter- and intra-paratope interactions

In the near absence of secondary structure, disulphide bonds appear to act as sources of stability for both peptides. For K92, both the backbone amide and carbonyl of $C23_{K92}$ participate in H-bonds with the side chain of $S82_{C5}$ (*Figure 3B*). For K8, an interchain sulphur–π stack between the $C27_{K8}$-$C41_{K8}$ disulphide bond and the aromatic of $Y1378_{C5}$ positions the hydroxyl group of $Y1378_{C5}$ to make a H-bond with $D25_{K8}$ (*Figure 3A*). While for K92, an intra-chain sulphur–π stack between the $C9_{K92}$-$C23_{K92}$ disulphide bond and the aromatic of $Y14_{K92}$ orientates $Y14_{K92}$, such that its hydroxyl group participates in an interchain H-bond with $N38_{C5}$.

### Comparison to known antibody paratopes

Although antibody-derived, K8 and K92 are structurally unique variable regions. We compared the K8 and K92 knob domains to a non-redundant set of 924 non-identical sequences of paired antibody–protein antigen structures from SAbDab (*Dunbar et al., 2014*). Paratopes were defined as any antibody residues within 4.5 Å of the antigen in the structure. The paratopes of K8 and K92 contain 18 and 10 residues, respectively, which are within the typical range of antibody paratope sizes (*Figure 3—figure supplement 4A*). Given this similarity in size, we searched for structurally and physicochemically similar antibody paratopes from the 924 antibody complexes but no similar paratope sites were found (*Wong et al., 2020*). While the limited examples preclude firm conclusions, this lack of similarity could be due to either the unusual fold of the knob domains or the differences in paratope amino acid composition.

In terms of residue usage, one difference in paratope composition that is potentially universal is the presence of cysteine in the knob domains (*Figure 3—figure supplement 4B*) which is uncommon in most antibody paratopes, with the exception of the CDR1-CDR3 disulphides, which have been described in camelid VHH (*Govaert et al., 2012*), and in broadly neutralising antibodies; those which cross react with several strains of a virus, and for which a disulphide bond in CDRH3 has been described in antibodies against HIV-1 (*Hutchinson et al., 2019*) and hepatitis C (*Flyak et al., 2018*). Using Arpeggio (*Jubb, 2015*) to identify inter- (antigen contacting) and intra-paratope interactions (hydrogen bond, polar, ionic, and hydrophobic) revealed that, on average, antibodies have 16 intra-paratope and 17 inter-paratope interactions; K8 is very close to this, with 15 intra-paratope and 17 inter-paratope interactions, whereas K92 paratope has fewer, with 9 intra-paratope and 10 inter-paratope interactions. A bovine Fab with an ul-CDRH3 was recently crystallised in complex with antigen, in this case a soluble portion of the HIV envelope (*Stanfield et al., 2020*). While the low resolution of the crystal structure hindered analysis, a casual inspection of the paratope suggests that 10 intra-paratope and 10 inter-paratope interactions are sustained by the knob domain, comparable to K92.

A search for structurally homologous proteins, using the DALI protein structure comparison server (*Holm, 2020*), did not find any 3D structures similar to K8 or K92, including the 14 known structures of bovine Fabs with ul-CDRH3 in the PDB. These results highlight the heterogeneity of these structural elements of the bovine immune system which likely arise through selection against a specific antigen/epitope. We next looked at homology with cyclic peptides. A recent review summarised the interactions mediated by cyclic peptides bound to proteins, across 65 co-crystal structures in the PDB (*Malde et al., 2019*). This revealed that cyclic peptides on average sustain eight electrostatic interactions with their protein target, with a range of 1–20. When we consider K8, its 19 inter-paratope interactions are comparatively high for a peptide, while the seven inter-paratope interactions of K92 are far more typical (*Figure 3—figure supplement 4C*).

### The structural basis for allosteric inhibition of C5 by K8 and K92

When compared to the binding sites of other C5 modulators (*Figure 4A*), it can be observed that the epitope for K92 is entirely contained within the binding interface of a previously reported immune evasion molecule, the 23 kDa SSL7 protein from *S. aureus* (*Figure 4B*). While the C5-SSL7 structure reveals a shallow binding site involving a series of five H-bonds between SSL7 and a region

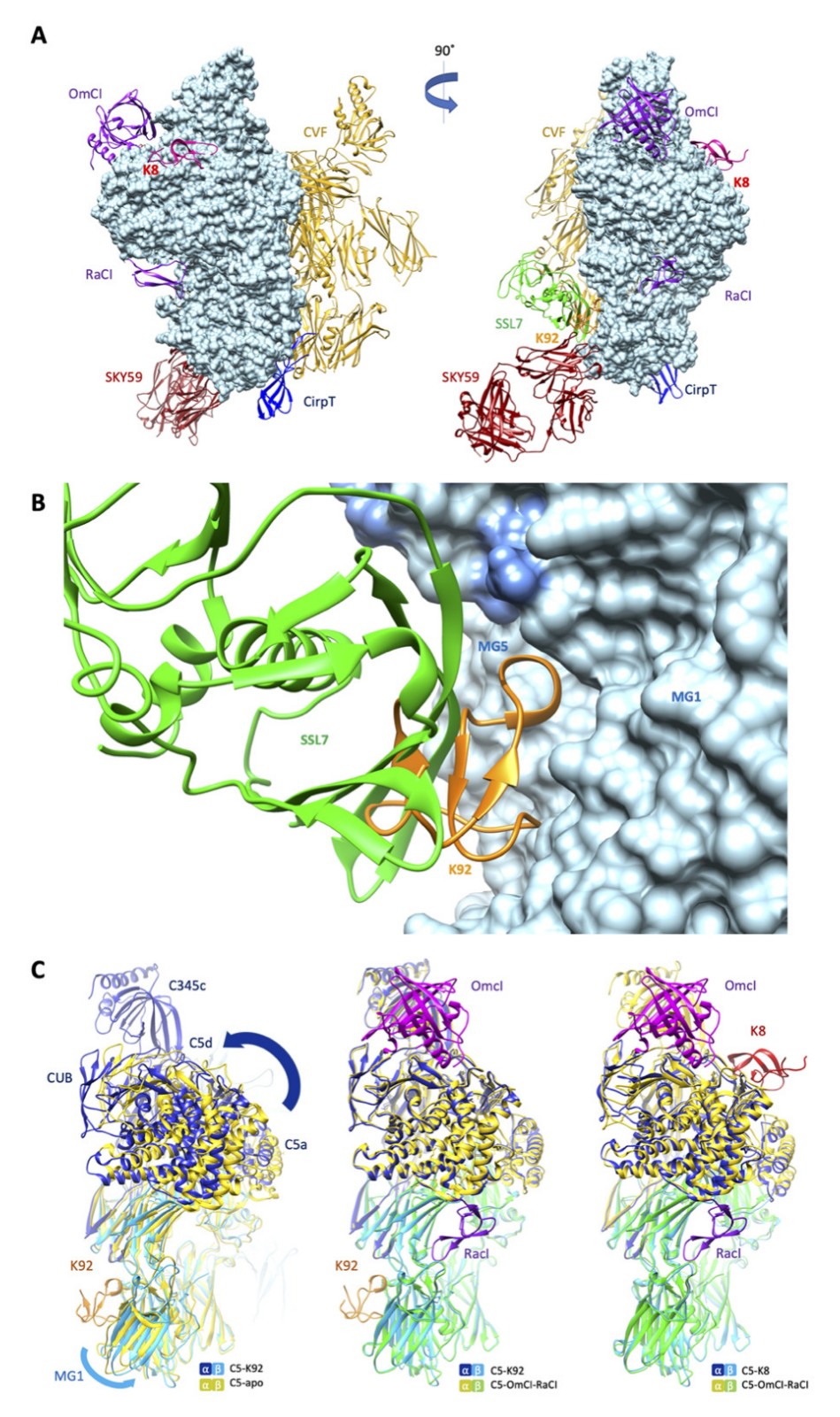

**Figure 4.** Comparison of the K8 and K92 binding sites with known C5 inhibitor complexes. Structural alignment of the complexes of C5 with the K8 and K92 knob domain peptides with the known structures for OmCI and RaCI (Protein Data Bank [PDB] accession code 5HCC; *Jore et al., 2016*), SSL7 and cobra venom factor (CVF) (PDB accession code 3PRX; *Laursen et al., 2011*), Cirp-T (PDB accession code 6RPT; *Reichhardt et al., 2020*), and SKY59 (PDB accession code 5B71; *Fukuzawa et al., 2017*) using UCSF Chimera (*Pettersen et al., 2004*). Alignments have been performed globally except for

*Figure 4 continued on next page*

*Figure 4 continued*

instances where the inhibitor has been crystallised bound to a single domain of C5. (**A**) shows two views of the superimposed C5-inhibitor complexes, differing by a 90° rotation. C5 is shown in molecular surface rendering, with ribbon representations of OmCI and RaCI in purple, SSL7 in green, CVF in gold, SKY59 in dark red, K8 in bright red, and K92 in orange. (**B**) shows a close-up view of the K92 binding site with that of SSL7 superimposed, for comparison. In contrast with the superficial binding mode of SSL7, K92 is wedged between the macroglobulin (MG)1 and MG5 domains of C5. (**C**) (left) shows that the interaction between K92 and C5 induces a slight separation of the MG1 and MG5 domains, resulting in a significant rotational movement of the C5a, C5d, and CUB domains, when compared to the C5-apo structure (PDB accession code 3CU7; *Fredslund et al., 2008*). (**C**) also shows that the complex with OmCI and RaCI (PDB accession code 5HCC; *Jore et al., 2016*) stabilises a similar conformation in C5 (**C**, middle) to that of K92, as well as K8 (**C**, right). For this structural comparison, the C5 MG5 domains of the complexes were superimposed.

The online version of this article includes the following figure supplement(s) for figure 4:

**Figure supplement 1.** Structural comparison of C5-K92 and C5-CVF complexes.

---

of β-sheet on the MG5 domain, spanning $H511_{C5}$-$E516_{C5}$ (*Laursen et al., 2010*), here we show that K92 is wedged between the MG1 and MG5 domains, inducing a re-orientation of the side chain of $H511_{C5}$ and forming a backbone H-bond with $F510_{C5}$. When comparing K92 and SSL7, the small changes observed in the binding pose achieve different allosteric effects; SSL7, either in isolation or in complex with its second ligand IgA, is full, or occasional partial, antagonist of both the AP and CP (*Bestebroer et al., 2010*; *Laursen et al., 2010*), while K92 is a selective partial antagonist of the AP.

Inspection of the C5a anaphylatoxin domain reveals that the C-terminus of the C5a domain in the C5-K92 complex adopts a helical conformation, which is analogous to the C5-OmCI-RaCI complex, burying the Bb-cleavage site (R751). In other C5 structures (including C5-apo and C5-CVF), this linker adopts an extended conformation following an unstructured loop and only sparse continuous electron density was observed for the linker extending from MG6 to C5a in the C5-K8 complex, possibly suggesting its R751 scissile bond is more exposed.

When the MG5 domains in the C5-K92 complex and the C5-apo structure are superimposed (*Figure 4C*), a slight twist can be observed in the MG1 domain, caused by the binding of K92 and resulting in a significant rotational movement of the C5 α-chain. A similar conformational change results from the binding of OmCI and RaCI, and to some extent K8, as these structures are virtually superimposable. CVF can form a highly stable C3/C5 convertase, following incubation with factor D and factor B in the presence of $Mg^{2+}$ (*Vogel and Müller-Eberhard, 1982*), which may offer a surrogate model for C5 convertase (*Laursen et al., 2011*). When superimposing C5-K92 and C5-CVF (PDB accession code 3PVM) complexes, C5 does not adopt a similar conformation as when bound by K92 and K8 (*Figure 4—figure supplement 1*), potentially indicating both knob domains stabilise a different C5 conformation than binds the C5 convertase.

When considering the organisation of the C5 convertases, the C5-CVF crystal structure reveals that CVF and C5 align perfectly to create a mirror image complex, with a conformational change in the C5 convertase site at arginine 751, potentially placing C5a within range of the catalytic unit of the MG7-associated convertase complex, offering a surrogate model for C5 convertase activation (*Laursen et al., 2011*). We have shown that K92 is not an orthosteric inhibitor of either the CP or the AP convertase, thereby precluding binding of the convertase to a cleft between the MG1–MG5 domains. As the K92 epitope is entirely contained within the SSL7 binding site, this is compatible with the CVF model for C5 activation, with a co-crystal structure of the ternary complex of C5, CVF, and SSL7 (PDB accession code 3PRX6), demonstrating that the CVF and SSL7 binding sites do not cross block. Also consistent with the CVF model for C5 activation, binding of K8 to the MG8 domain would not appear to sterically block the catalytic unit. We therefore sought to further explore the apparent conformational changes in our structures.

## Solution techniques reveal allosteric networks

To validate the apparent conformational changes occurring in C5 due to the binding of K8 and K92 as revealed by our crystal structures, we analysed the C5-knob domain complexes by two-solution biophysical techniques – small-angle X-ray scattering (SAXS) and hydrogen-deuterium exchange mass spectrometry (HDX-MS).

SEC-SAXS, where size exclusion chromatography (SEC) immediately precedes the solution X-ray experiment ensuring a monodispersed sample, was performed in concert with SEC multi-angle laser

light scattering (SEC-MALLS). Data were collected for C5 and the C5-K8, C5-K57, C5-K92, and C5-K149 complexes (*Figure 5A–C*). SEC-MALLS confirmed that the increases in molecular weight of the complexes were consistent throughout the elution peaks (*Supplementary file 1,* Table 3.1 and *Figure 5—figure supplement 1A*). While SEC-SAXS elution profiles gave stable estimates of the radius of gyration ($R_G$) across the tip of the peak, frames (scattering curves collected during the elution of the sample) from the descending elution peaks show lower $R_G$ values, suggesting the presence of unbound C5.

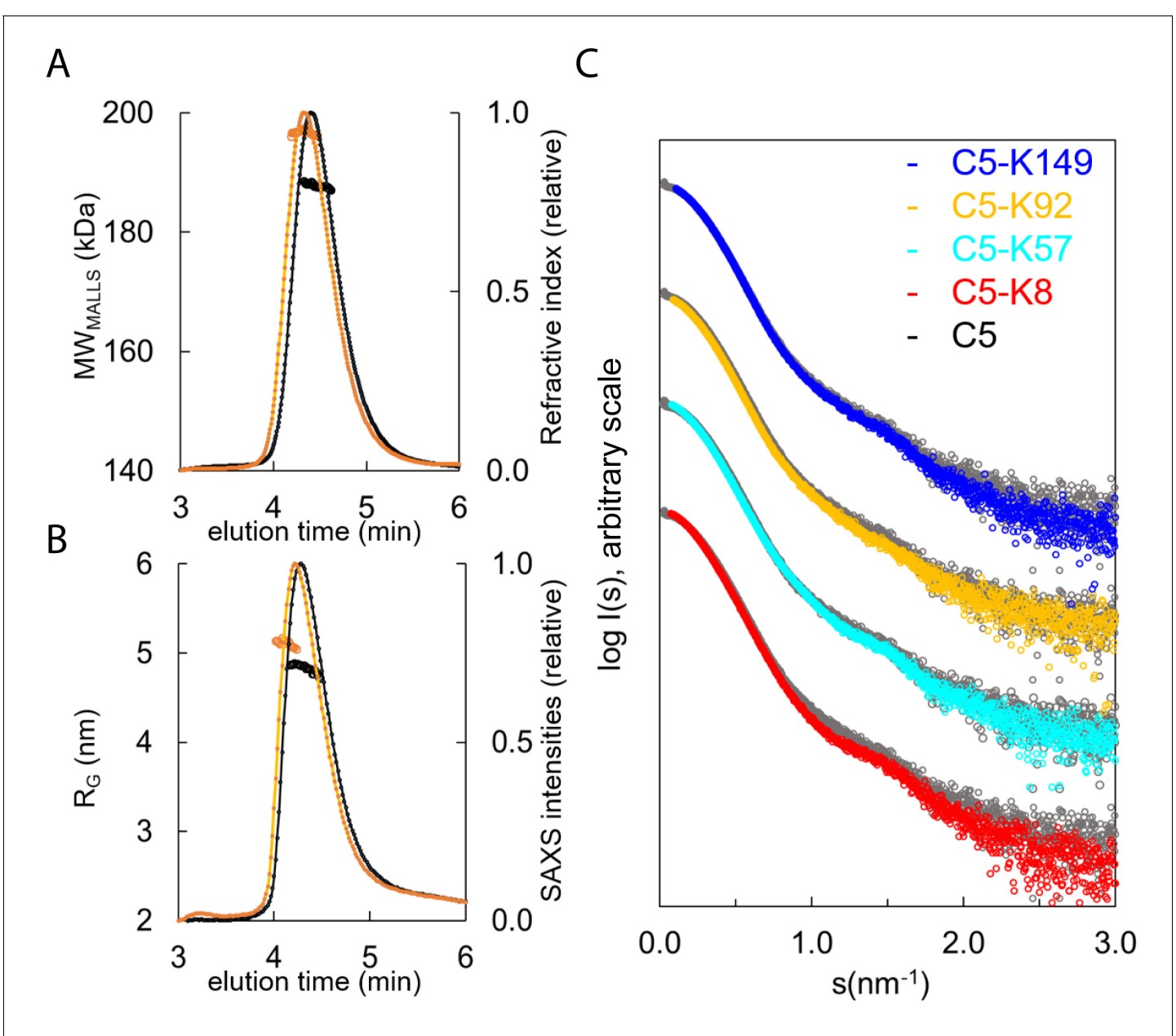

**Figure 5.** Hydrodynamic properties and solution conformation of C5 and C5-knob domain complexes by small-angle X-ray scattering (SAXS). Size exclusion chromatography multi-angle laser light scattering (SEC-MALLS) chromatograms (**A**) for apo C5 (black) and C5-K92 (orange) show a homogenous molecular weight increase across the C5-K92 elution peak. The SEC-SAXS elution profile collected under identical experimental conditions (**B**) shows an increase in radius of gyration ($R_G$) for the C5-K92 complex. Scattering curves of all C5-knob domains are shown (**C**); the C5-knob domain complexes are shown against apo C5 (in grey), and for ease of viewing, the curves are arbitrarily shifted in the Y axis.
The online version of this article includes the following figure supplement(s) for figure 5:

**Figure supplement 1.** SAXS analyses of the C5-knob domain peptide complexes.

Frames corresponding to the tip of the peak were averaged and submitted for full SAXS analysis. For the complexes, the scattering curves showed slight increases in both the $R_G$ and solute volume (*Supplementary file 1,* Table 3.1), with the C5-K8 complex showing the largest change and C5-K149 the smallest change, corresponding with the absence of function and suggesting K149 binds peripherally to a conformation closely resembling C5-apo. For K92 and K57, the discrepancies observed in the mid s range indicate an overall change in flexibility of C5 upon binding of these peptides, and this tuning of dynamics may contribute to their mechanism.

Consistent with earlier observations (*Fredslund et al., 2008*), comparison of the C5-apo experimental data with the theoretical scattering curve revealed discrepancies in the lowest angle range, indicating C5 adopts a more elongated conformation in solution than the crystal structure would suggest (*Figure 5—figure supplement 1B*). To better approximate C5 in solution, we performed a normal mode model analysis (NMA) using SREFLEX (*Panjkovich and Svergun, 2016*) and found that elongation of the C5 model improved the $\chi^2$ from >13 to 1.55. The fit of the C5-K92 complex was also markedly improved by the NMA, whereby elongation and incorporation of the peptide improved the model from an initial $\chi^2$ of >20, to 2.5 (with an overall root mean square [RMS] of 3.8 in both cases).

When using the C5-K8 co-crystal structure for fitting of the C5-K8 SAXS data, the absence of the C345c domain was problematic. The generation of a hybrid model where the C345c domain was reinstated initially produced a poor fit ($\chi^2$=75). A restrained rigid body analysis of this model followed by NMA refinement allowed us to significantly reduce the discrepancy to $\chi^2$=4.1 indicating an overall acceptable fit. The $\chi^2$ value is still somewhat larger than those observed for the other complexes, which may suggest an increased flexibility around the C345c linker. This result correlates with the absence of clear electron density for the C345c domain in the crystal structure. The latter may be a consequence of K8 inducing additional flexibility to this region, which again could contribute to the efficacy of the peptide.

The discrepancies between the crystal structures and the solution scattering data indicate that while permitting elucidation of the molecular interaction of the epitopes, the constraints of the crystal lattice may impede the detection of more subtle, global changes, leading to underestimation of the conformational changes induced by the peptide.

To further explore such effects in solution, we used HDX-MS to provide molecular-level information on local protein structure and dynamics. HDX-MS measures the exchange of backbone amide hydrogen to deuterium in the solvent, with the rate of HDX determined by solvent accessibility, protein flexibility, and hydrogen bonding. To interpret the impact of peptide binding on C5 structural dynamics, we performed differential HDX (ΔHDX) analysis, comparing C5-knob domain complexes to apo C5, where shielding of C5 residues through participation in a binding interface will prevent deuteration, while conformational changes may increase or decrease deuterium uptake, in relation to the degree of solvent exposure.

For C5-K8, the sole protected region of C5 corresponded to the epitope on the MG8 domain ($L1380_{C5}$-$E1387_{C5}$), although the interface was not entirely defined (*Figure 6A*, see also *Figure 6—figure supplement 1* and *Supplementary file 1,* Table 3.2). Additional conformational changes were observed in the neighbouring C5d domain which becomes more solvent exposed, suggesting K8 is affecting the dynamics of this domain.

For the C5-K92 complex, consistent with the crystal structure, there was protection of the C5 residues located in the epitope between the MG1 and MG5 domains ($H70_{C5}$-$L85_{C5}$), shown in *Figure 6C*. There were also effects distal to the K92 binding site, notably in C5d ($I1169_{C5}$-$F1227_{C5}$) and neighbouring CUB domain ($L1303_{C5}$-$L1346_{C5}$), indicating a K92-induced conformational change. Interestingly, the allosteric network can be visualised by changes in solvent exposure which propagate from the K92 epitope through MG2 domain ($L126_{C5}$-$V145_{C5}$) and into the C5d and CUB domains. For the C5-K57 complex, the absence of a co-crystal structure meant we had no prior knowledge of the K57 epitope. However, clear protection was observed in the MG5 domain, immediately adjacent to the K92 epitope ($N483_{C5}$-$L540_{C5}$), with sparse areas of increased solvent exposure located in the MG6 ($Q572_{C5}$-$L590_{C5}$), MG8 ($L1379_{C5}$-$A1388_{C5}$), and C5d ($K1048_{C5}$-$Y1064_{C5}$) domains (*Figure 6B*). A single protected peptide was also present in the CUB domain ($G951_{C5}$-$L967_{C5}$), suggesting the K57 epitope may be on either the MG5 or CUB domains.

There was little protection or deprotection of proteolytic fragments of the C5a domain in any of the complexes; we therefore propose that the knob domain peptides do not act by inducing

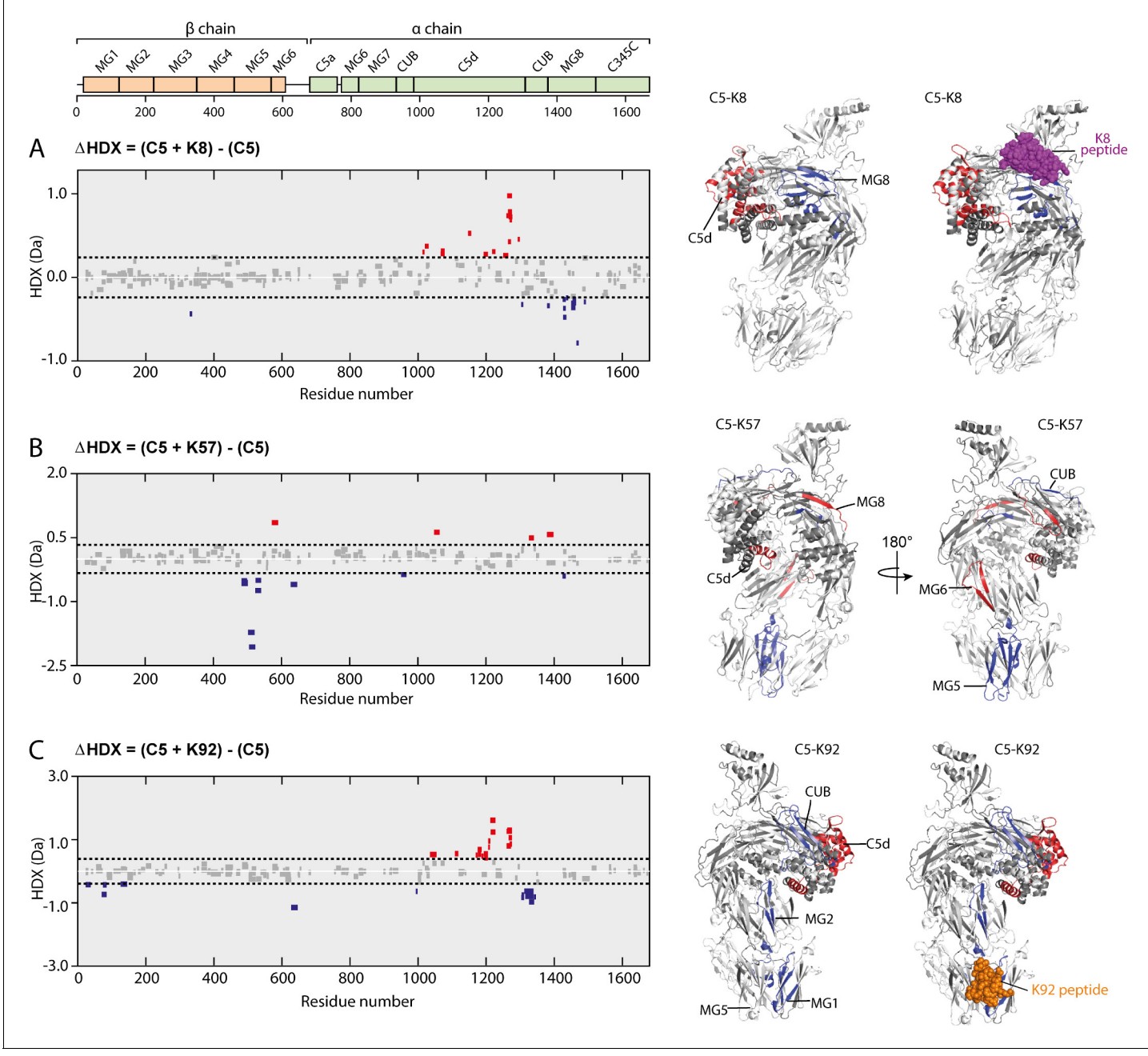

**Figure 6.** Impact of knob domain binding on the structural dynamics and conformation of C5. Differential hydrogen-deuterium exchange (ΔHDX) plots for C5 in complex with knob domains (**A**) K8, (**B**) K57, and (**C**) K92 at 1 hr of deuterium exposure. Blue denotes peptides with decreased HDX (backbone H-bond stabilisation), and red denotes peptides with increased HDX (backbone H-bond destabilisation). 98% confidence intervals are shown as dotted lines. Peptides in grey have insignificant ΔHDX. Measurements were performed in triplicate, and all HDX-MS peptide data are detailed in *Supplementary file 1* Table 3.2. ΔHDX for C5 + K8, C5 + K57, and C5 + K92 are coloured onto C5 (Protein Data Bank accession code 5HCC, minus OmCI and RaCI).

The online version of this article includes the following figure supplement(s) for figure 6:

**Figure supplement 1.** HDX analyses of the C5-knob domain peptide complexes.

conformational changes which shield the scissile arginine bond. Although in the structure of the C5-K92 complex the Bb-cleavage site is more buried compared to that in the C5-K8 complex. Taken in the context of the other changes, notably in the C5d and CUB domains, it is more probable that they affect more global changes in C5 which lower the affinity for C3b or the C5 convertases. The

HDX-MS data are in good agreement with our crystallographic data, with the K8 and K92 epitopes defined as clear areas of solvent protection. The conformational change in the C5d domain and significant rotational moment of the C5 α-chain, which were evident upon alignment of the MG1 domain of apo C5 with the C5-K8 and C5-K92 co-crystal structures, also appears to manifest in solution in response to binding of the knob domains.

### Putative K57 binding site

To further home in on the K57 binding site, we measured binding to C5b and C5b-6 in an SPR single-cycle kinetics experiment (*Supplementary file 1,* Table 4.1). Upon cleavage of C5a, the remaining domains of the α-chain undergo a substantial conformational change, mediated by rearrangement of the MG8, CUB, and C5d domains (*Hadders et al., 2012*; *Aleshin et al., 2012*). The resulting C5b subunit is metastable and prone to aggregation and decay, which leaves it unable to bind C6 or form the MAC. By SPR, K8 did not bind C5b but could C5b-6. However, K57, K92, and K149 all bound C5b and C5b-6 (*Supplementary file 1* Table 4.2). For C5b, this was within two-fold of their previously published affinities for C5 (17.8 nM for K8, 1.4 nM for K57, and <0.6 nM for K92; *Macpherson et al., 2020*), except for K149, which displayed threefold higher affinity for both C5b and C5b-6 than previously reported for C5 (15.5 nM; *Macpherson et al., 2020*). K92 did exhibit lower affinity for C5b-6 complex than C5, binding the complex at 6.7 nM, relative to <0.6 nM for C5 alone. As the CUB domain is significantly altered in C5b, this increases the likelihood that, of the two protected regions identified by HDX-MS, the K57 epitope is on the MG5 domain.

## Discussion

We present a new family of peptides, bovine antibody-derived knob domains, and show that, upon binding of the therapeutic target complement component C5 with high affinity, they display a range of inhibitory mechanisms. Knob domain peptides rely on the bovine immune system to achieve high-affinity binding through optimisation of amino acid composition, 3D structure, and disulphide bond network. They have only been recently isolated as a practicable antibody fragment (*Macpherson et al., 2020*), and therefore, structural and functional analysis of the complexes with their clinical targets will greatly aid their development as therapeutics.

### Function

Functional characterisation at the level of individual complement pathways identified K57 as a novel C5 inhibitor, which is a fully efficacious inhibitor of the terminal pathway in response to both CP and AP activation, and a potential therapeutic candidate for complement-mediated disorders, such as paroxysmal nocturnal haemoglobinuria and atypical haemolytic uraemic syndrome. Additionally, the discovery of K149 as a 'silent binder' of C5 may be of considerable value as a non-inhibitory reagent for the detection of native C5 (*Figure 1*). K8 and K92 both displayed allosteric inhibitory activity against C5. K92 achieved selective inhibition of the AP through a non-competitive mechanism (*Figure 1*). To our knowledge, this is the first reported example of complement pathway-specific inhibition through C5 and the first experimental evidence reported for mechanistic differences between the AP and CP C5 convertases. This suggests an expanded therapeutic scope for C5, whereby tuning of the conformational ensemble or dynamics with allosteric compounds can bias activation to leave certain complement pathways intact. Complete inhibition of the terminal pathway has been shown to increase the susceptibility of eculizumab patients to *Neisseria meningitidis* infections (*McNamara et al., 2017*). Selective inhibition of C5-cleavage by the AP C5-convertase, and not the CP C5-convertase, may partially preserve serum bactericidal activity, thereby lowering the risk of meningococcal disease.

### Structure

Structural analyses, utilising X-ray crystallography, revealed the unique topologies of knob domain peptides K8 and K92 and their distinctive binding modes in C5. Due to the apparent structural homology of knob domains with certain venomous peptides, of which conotoxins and spider venoms are examples, it has been proposed that the knob domains of ul-CDRH3 might be similarly predisposed to target the concave epitopes of ion channels. Likewise, structural homology with defensin peptides has garnered hypotheses regarding an improved ability to bind viral capsid coats. Indeed,

bovine antibodies with ul-CDRH3 have been raised against the viral capsid of HIV with exceptional efficiency, given the challenging nature of the antigen (*Sok et al., 2017*; *Stanfield et al., 2020*). However, the study presented here shows that, in the case of C5, concave epitopes are not the knob domain's sole preserve. Notably, the MG8 domain epitope of K8 offers a planar pharmacophore and, while the K92 epitope is more undulating, casual inspection of the C5 structure reveals numerous deeper cavities available (*Figure 3*). This may mean that knob domains can be raised to inhibit flat surfaces involved in protein–protein interactions, which might not offer binding sites for orthosteric small molecules.

We note that the structural architecture of the knob domains varies for the epitope. Their immune derivation means that, unlike cysteine-rich peptides derived from other natural sources, such as venoms, the bovine immune system can be used to define specificity for any antigen. Comparative structural analysis suggests that knob domain paratopes are differentiated from conventional antibodies and the structures of known cyclic peptides, offering a different binding architecture to other small antibody fragments, such as the camelid VHH. While firm conclusions are hampered by limited examples, the number of interactions does not seem dissimilar from cyclic peptides or mAbs, both of which have been successfully applied to tackling high-affinity protein–protein interactions.

Our structures demonstrate that the importance of the network of disulphide bonds goes beyond a stabilising role. An apparent paucity of secondary structure would suggest that while stabilisation of the domain is indeed critical, disulphide bonds also participate in sulphur–π interactions to sustain intra- and inter-chain interactions. The structures of knob domains bound to their target antigen demonstrate both the diversity and versatility afforded to the bovine immune repertoire by these sequences.

## Mechanism

Structural alignment of the C5-K92 co-crystal structure with the apo C5 structure (*Figure 4*), using the MG5 domain, revealed a rotational movement of the MG1 causing the α-chain to adopt a twisted conformation accompanied by a rotational movement in the C5d domain, in response to knob domain binding. Comparison of the C5-K8 structure with the apo C5 structure revealed a similar conformation but with less movement in the C5d domain. The helical C5d domain is the target of two immune evasion molecules which have evolved in ticks, OmCI and RaCI, both of which inhibit C5 by crosslinking C5d to neighbouring domains (*Jore et al., 2016*). Additionally, it has been shown that polyclonal antibodies raised against C5d inhibit binding of C5 to C3b (*DiScipio, 1981*). The binding site of OmCI is contained within the CUB and C5d domains, with only a single, non-bonded interaction to the C345c domain visible in the crystal structure (*Jore et al., 2016*), which appears mediated by crystal contacts. Interestingly, the C5-OmCI-RaCI crystal structure (PDB accession code 5HCC) reveals similar conformational changes in C5d, relative to the apo C5 structure. We therefore propose that rearrangement of C5d can lower the affinity, or preclude the interaction, of C5 for the convertases and that this may be a common inhibitory mechanism for OmCI, RaCI, K8, and K92. Should K92 and K8 inhibit C5 by modulating the C5d domain, in the case of K92, this occurs at a range of over 50 Å. Such remote effects are not unprecedented; allosteric structural changes can be propagated at over 150 Å in response to drug binding (*Haselbach et al., 2017*).

Subsequent solution biophysics methods substantiate our crystallographic observations. HDX-MS analysis revealed areas of solvent protection changes in the MG8 domain, resulting from the binding of K8, and in the MG1 and MG5 domains of the C5-K92 complex (*Figure 6*), corresponding to their respective epitopes, as identified with X-ray crystallography (*Figure 3*) and confirmed by site-directed mutagenesis analysis. Changes in solvent exposure were also observed in the α-chain for C5-K92, providing a route to visualise the allosteric network. As similar conformational or dynamic changes occur both in solution and in the crystal structure, this suggests that the effects are ligand induced and are not the result of crystal packing interactions. SAXS analysis also suggests that K8 and K92 increase the flexibility of C5 and effects on dynamics may be a contributing factor in realising efficacy (*Figure 5*).

For K57, which had an $E_{max}$ of 100% for both pathways and was not demonstrably allosteric, HDX-MS and biacore experiments with the metastable C5b suggested a putative epitope on the MG5 domain. This could support an orthosteric mechanism of action as CVF, which can form a stable C5 convertase, contacts the MG5 domain in the C5-CVF co-crystal structure (PDB accession code 3PVM). However, by SAXS, similar changes in conformation and/or dynamics to the C5-K8 and C5-

K92 complexes were apparent and an allosteric network, including changes in the C5d domain, was observed in the C5-K92 complex by HDX-MS. These observations could support an allosteric mechanism for K57.

Importantly, we also saw a high degree of negative cooperativity between the different knob domains by SPR (*Figure 2*), suggesting that all the functional knob domains perturb the conformational state or dynamics of C5. K57 also showed cooperativity with other functional knob domains, suggesting providing further evidence that it stabilises a conformation of C5 that is less energetically favourable for binding of the other ligands.

Our observations with K92 suggest that further work may be required to elucidate the mechanism of action of another binder of the MG1 and MG5 domains, SSL7. Given that SSL7 can be a partial inhibitor (*Laursen et al., 2010*), even with co-binding of IgA, this precludes a steric mechanism and invites biophysical studies in solution. Additionally, another tick-derived inhibitor, Cirp-T, was also recently reported as predominantly binding to the MG4 domain, with an orthosteric mechanism of action attributed. However, we note that published data only showed an $E_{max}$ of <90% in AP-driven assays (*Reichhardt et al., 2020*), indicating that it is an allosteric C5 inhibitor for the AP, and potentially also the CP, C5 convertase, which may merit further investigation.

In conclusion, we introduce knob domains as a new peptide modality, with unexplored therapeutic potential for the modulation of proteins and protein–protein interactions. This study is the first application of knob domain peptides and reveals an unexpectedly high incidence of allosteric modulators of complement C5, expanding its scope for complement-targeted therapies and providing important mechanistic tools for the study of C5 convertases. Knob domains can offer a range of advantages over the current macromolecular C5 inhibitors, including their use as peptide therapeutics, while grafting knob domains into the CDRH3 of well-characterised Fabs or using Fc tags could provide routes to extend half-life in vivo by attenuating renal clearance.

# Materials and methods

## Complement proteins

Human C5 was affinity purified using an E141A, H164A OmCI column (*Macpherson et al., 2018*). Briefly, human serum (TCS Biosciences, Botolph Claydon, UK) was diluted 1:1 (v/v) with phosphate buffered saline (PBS), 20 mM ethylenediaminetetraacetic acid (EDTA), and applied to a 5 mL Hi-Trap NHS column (GE Healthcare, Amersham UK), which contained 20 mg of E141A H164A OmCI protein, at a rate of 1 mL/min. The column was washed with 5× column volumes (CV) of PBS, C5 was then eluted using 2 M $MgCl_2$ and immediately dialysed into PBS. C5b was prepared from human C5 by incubating C5 with CVF, factor B and factor D, at a 1:10 molar ratio, as previously described (*Jore et al., 2016*). C5a was removed using a spin column with 30 kDa cut-off (Thermo Fisher Scientific, Horsham, UK).

## Knob domain peptide production

Knob domain peptides were expressed fused to the CDRH3 of the PGT-121 Fab, as previously described (*Macpherson et al., 2020*). Plasmid DNA for each construct was amplified using QIAGEN Plasmid Plus Giga Kits. Expi293F cultures were transfected with Expifectamine 293 Transfection kits (Invitrogen, Renfrew, UK) as per the manufacturer's instructions. The cells were cultured for 4 days and supernatants harvested by centrifugation at 4000 rpm for 1 hr. Harvested supernatants were applied to a Hi-Trap Nickel excel columns (GE Healthcare, Amersham, UK) using an Akta pure (GE Healthcare, Amersham, UK). Cell supernatants were loaded at 2.5 mL/min, followed by a wash of 7× CV of PBS, 0.5 M NaCl. A second wash with 7× CV of buffer A (0.5 M NaCl, 0.02 M Imidazole, PBS pH 7.3) was performed, and samples were eluted by isocratic elution with 10× CV of buffer B (0.5 M NaCl, 0.25 M) Imidazole, PBS (pH 7.3). Post elution, the protein-containing fractions were pooled and buffer exchanged into PBS using dialysis cassettes (Thermo Fisher Scientific, Horsham, UK).

For isolation of the knob domain peptide, PGT-121 Fab-knob peptide fusion proteins were incubated with tobacco etch virus protease, at a ratio of 100:1 (w/w), for a minimum of 2 hr at room temperature. Peptides were purified using a Waters UV-directed FractionLynx system with a Waters XBridge Protein BEH C4 OBD Prep Column (300 Å, 5 μm, 19 × 100 mm, Waters Corp., Milford, MA). An aqueous solvent of water, 0.1% trifluoroacetic acid (TFA), and an organic solvent of 100%

MeCN was used. The column was run at 20 mL/min at 40°C with a gradient of 5–50% organic solvent, over 11 min. Fractions containing knob peptide were pooled and lyophilised using a Labconco Freezone freeze drier.

## Complement activation assays

For the C3 and C9 ELISAs, microtiter plates (MaxiSorp; Nunc) were incubated overnight at 4°C with 50 µL of a solution of in 75 mM sodium carbonate (pH 9.6) containing either 2.5 µg/mL aggregated human IgG (Sigma-Aldrich, Gillingham, UK) for CP or 20 µg/mL zymosan (Sigma-Aldrich, Gillingham, UK) for AP. As a negative control, wells were coated with 1% (w/v) bovine serum albumin (BSA)/PBS. Microtiter plates were washed four times with 250 µL of wash buffer (50 mM Tris-HCl), 150 mM NaCl and 0.1% Tween 20 (pH 8) between each step of the procedure. Wells were blocked using 250 µL of 1% (w/v) BSA/PBS for 2 hr at room temperature. Normal human serum was diluted in either gelatin veronal buffer with calcium and magnesium (GVB$^{++}$: 0.1% gelatin, 5 mM Veronal buffer, 145 mM NaCl, 0.025% NaN$_3$, 0.15 mM calcium chloride, 1 mM magnesium chloride, pH 7.3; for CP) or Mg-ethylene glycol tetraacetic acid (Mg-EGTA) (2.5 mM veronal buffer [pH 7.3] containing 70 mM of NaCl, 140 mM of glucose, 0.1% gelatin, 7 mM of MgCl2, and 10 mM of EGTA; for AP). Serum was used at a concentration of 1% in CP or 5% in AP and was mixed with serially diluted concentrations of peptides (16 µM – 15.6 nM) in GVB$^{++}$ or Mg-EGTA buffer, and preincubated on ice for 30 min. Peptide–serum solutions were then incubated in the wells of microtiter plates for 35 min for CP assays (both C3b and C9 detection) or 35 min for AP (C3b) or 60 min for AP (C9), at 37°C. Complement activation was assessed through detection of deposited complement activation factors using specific antibodies against C3b (rat anti-human C3d HM2198, Hycult, Uden, The Netherlands) and C9 (goat anti-human C9, A226, Complement Technologies Tyler, TX) at a 1:1000 dilution. Bound primary antibodies were detected with horse rdischHRP-conjugated goat anti-rat (ab97057, Abcam, Cambridge, UK) or rabbit anti-goat (P0449, Agilent Dako, Santa Clara, CA) secondary antibodies at a 1:1000 dilution. Bound HRP-conjugated antibodies were detected using TMB One solution (Eco-TEK – manufactured by Bio-TEK, Winooski, VT) with absorbance measured at 450 nm.

For the C5b ELISA, assays were run using the CP and AP Complement functional ELISA kits (Svar Life Science, Malmö, Sweden). For sample preparation, serum was diluted as per the respective protocol for the CP and AP assays. Serial dilutions of peptides were prepared and allowed to incubate with serum for 15 min at room temperature prior to plating.

For the C5a ELISA, assays were run using the Complement C5a Human ELISA Kit (Invitrogen, Renfrew, UK). For sample preparation, at the end of the 37°C incubation of the serum/peptide samples on the C5b ELISA plate, 50 µL of the diluted, activated serum was transferred to a C5a ELISA plate containing 50 µL/well of assay buffer. All subsequent experimental steps were performed as described in the protocol.

## Haemolysis assays

GVB$^{++}$ or Mg EGTA buffers, which had been supplemented with 2.5% glucose (w/v), were used for the CP and AP assays, respectively. For the AP, 150 µL of rabbit erythrocytes (TCS Biosciences, Botolph Claydon, UK) were washed twice, by addition of 1 mL of buffer and centrifugation at 800 $\times g$ for 1 min, and finally resuspended in 500 µL of buffer. For the CP, 150 µL sheep erythrocytes (TCS Biosciences, Botolph Claydon, UK) were washed twice with 1 mL of buffer and sensitised with a 1/1000 dilution of rabbit anti-sheep red blood cell stroma antibody (S1389, Sigma Aldrich, Gillingham, UK). After a 30°C/30 min incubation, with shaking, the cells were rewashed and resuspended with 500 µL of buffer. Serial dilutions of peptide were prepared in the respective buffers and normal human serum was added at 1% for the CP and 4.5% for the AP (corresponding to CH50 of the serum). Also, 90 µL of peptide–serum mixtures were plated into a V-bottom 96-well microtiter plate (Corning) and 10 µL of erythrocytes were added. Plates were incubated for 30 min at 37°C, with shaking. Finally, 50 µL of buffer was added, the plates centrifuged at 800 $\times g$, and 80 µL of supernatant was transferred to an ELISA plate (Nunc) and absorbance measured at 405 nm.

## Crystallography and structure determination

6.1 mg/mL C5 (20 mM Tris-HCl, 75 mM NaCl, pH 7.35) was mixed at a 1:1 molar ratio with either the K8 or K92 peptides. Crystallisation trials were initiated by the vapor diffusion method at 18°C with a 1:1 mixture of mother liquor (v/v). C5-K8 crystals were grown in a mother liquor of 0.1 M N-(2-acetamido)iminodiacetic A, 14% ethanol (v/v), pH 6.0. For C5-K92 crystals, the mother liquor was 0.1 M bicine/Trizma (pH 8.5), 10% (w/v) polyethylene glycol 8000, 20% (v/v) ethylene glycol, 30 mM sodium fluoride, 30 mM sodium bromide, and 30 mM sodium iodide (*Gorrec, 2009*). Prior to flash freezing in liquid nitrogen, C5-K8 crystals were cryoprotected in mother liquor with 30% 2-methyl-2,4-pentanediol (v/v). C5-K92 crystals were frozen without additional cryoprotection.

Data were collected at the Diamond Light Source (Harwell, UK), on beamline I03, at a wavelength of 0.9762 Å. The C5-K8 structure was solved using the automated molecular replacement pipeline Balbes (*Long et al., 2008*) using the apo C5 structure (PDB accession code 3CU7), minus the C345c domain. The C5-K8 complex crystallised in space group P2$_1$2$_1$2$_1$ with one molecule in the asymmetric unit. A backbone model of the K8 peptide was produced using ARP-wARP (*Langer et al., 2008*) which informed manual model building in Coot (*Emsley et al., 2010*), within the CCP4 suite (*Winn et al., 2011*). The model was subjected to multiple rounds of refinement in Refmac (*Murshudov et al., 1997*) and Phenix (*Adams et al., 2010*). The overall geometry in the final structure of the C5-K8 complex is good, with 97.2% of residues in favoured regions of the Ramachandran plot and no outliers.

The C5-K92 complex crystallised in space group C2 with one molecule in the asymmetric unit. C5 was solved by molecular replacement with Phaser (*McCoy et al., 2007*) using the C5-OmCI-RaCI structure (PDB accession code 5HCC), with OmCI and RaCI removed. Manual building of the K92 peptide in Coot was greatly informed by mass spectroscopy disulphide mapping experiments. The model was subjected to multiple rounds of manual rebuilding in Coot and refinement in Phenix (*Adams et al., 2010*). The overall geometry in the final structure of the C5-K92 complex is good, with 95.2% of residues in favoured regions of the Ramachandran plot and no outliers. Structure factors and coordinates for both C5-knob domain peptide complexes have been deposited in the PDB (PDB accession codes: 7AD6 (C5-K92 complex) and 7AD7 (C5-K8 complex)). Crystal trials were also performed with the C5-K57 and C5-K149 complexes, but the resulting crystals diffracted poorly.

## Disulphide mapping of K92 peptide

A 250 µL K92 peptide at 1 mg/mL was alkylated with addition of 18 µL of 2-Iodoacetamide (Thermo Fisher Scientific, Horsham, UK) at room temperature for 30 min. Overnight dialysis into assay buffer (7.5 mM Tris-HCl, 1.5 mM CaCl$_2$, pH 7.9) was performed using 2 kDa slide-a-lyzer cassettes (Thermo Fisher Scientific, Horsham, UK). Chymotrypsin (sequencing grade, Roche Applied Sciences) was reconstituted to 1 µg/µL in assay buffer, and 5 µL of reconstituted enzyme was added to 80 µL of sample. Once mixed, the sample was incubated at 37°C for 1.5 hr before being quenched with 5 µL of 1% TFA. Samples were diluted 1 in 10 and 5 µL was loaded onto the analytical column.

Liquid chromatography electrospray ionisation mass spectrometry was acquired using an Ultimate 3000 UHPLC system (Thermo Fisher Scientific, Horsham, UK) coupled with a Q-Exactive Plus Orbitrap (Thermo Fisher Scientific, Horsham, UK). Separations were performed using gradient elution (A: 0.1% formic acid; B: 0.1% formic acid in acetonitrile) on an Acquity UPLC BEH C18 column (130 Å, 1.7 µm, 2.1 × 150 mm; Waters Corp., Milford, MA) with the column temperature maintained at 40°C.

The following analytical gradient at 0.2 mL/min was used: 1% B was held for 2 min, 1–36% B over 28 min, 36–50% over 5 min, and 50–99% B over 0.5 min. There were sequential wash steps with changes in gradient of 99%–1% B over 0.5 min (at a higher flow rate of 0.5 mL/min) before equilibration at 1% B for 6.5 min (at the original 0.2 mL/min).

A full MS/dd-MS2 (Top 5) scan was run in positive mode. Full MS: scan range was 200–2000 m/z with 70,000 resolution (at 200 m/z) and a 3 × 10$^6$ AGC target (the maximum target capacity of the C-trap), 100 ms maximum Injection time. The dd-MS2: 2.0 m/z isolation window, CID fragmentation (NCE 28) with fixed first mass of 140.0 m/z, with a 17,500 resolution (at 200 m/z), 1 × 10$^5$ AGC target, 200 ms maximum injection time. The source conditions of the MS were capillary voltage, 3 kV;

S-lens RF level, 50; sheath gas flow rate, 25; auxiliary gas flow rate, 10; auxiliary gas heater temperature, 150°C; and the MS inlet capillary was maintained at 320°C.

Data were acquired using XcaliburTM 4.0 software (Thermo Fisher Scientific, Horsham, UK), and raw files were analysed by peptide mapping analysis using Biopharma Finder 2.0 software (Thermo Fisher Scientific, Horsham, UK) by performing a disulphide bond search with a chymotrypsin (medium specificity) digest against the K92 peptide sequence. Assignments and integrations from Biopharma Finder were filtered to include only peptides identified as containing a single disulphide bond and with an experimental mass within |5| ppm of the theoretical mass. Intensities for all peptides containing the same cysteines pairing were summed and percentages were obtained from the summed against total intensities.

## SPR multicycle kinetics

Kinetics were measured using a Biacore 8K (GE Healthcare, Amersham, UK) with a CM5 chip, which was prepared as follows: 1-ethyl-3-(−3-dimethylaminopropyl) carbodiimide hydrochloride (EDC)/N-hydroxysuccinimide (NHS) was mixed at 1:1 ratio (flow rate, 10 μL/min; contact time, 30 s), and human C5 at 1 μg/mL in pH 4.5 sodium acetate buffer was injected over flow cell one only (flow rate, 10 μL/min; contact time, 60 s). Final immobilisation levels in the range of 2000–3000 RUs were obtained to yield theoretical Rmax values of ~50–60 RU. Serial dilutions of K8 and K92 knob domains, and various mutants, were prepared in HBS-EP (0.01 M HEPES pH 7.4, 0.15 M NaCl, 3 mM EDTA, 0.005% v/v Surfactant P20) buffer and injected (flow rate, 30 μL/min; contact time, 240 s; dissociation time, 6000 s). After each injection, the surface was regenerated with two sequential injections of 2 M $MgCl_2$ (flow rate, 30 μL/min; contact time, 30 s). Binding to the reference surface was subtracted, and the data were fitted to a single-site binding model using Biacore evaluation software.

## Binding pose metadynamics

Simulation structures were prepared using Schrodinger's Maestro Protein preparation wizard. The molecular dynamics runs were performed using the Schrodinger's default implementation of the binding pose metadynamics with the peptide chain considered in place of a ligand. Additional RMSD calculations for the peptide internal structure assessment in the last 20% of the dynamics were performed relative to the starting structures.

## Small-angle X-ray scattering

Data was collected at the EMBL P12 beam line (PETRA III, DESY Hamburg, Germany; *Blanchet et al., 2015*). Data was collected with inline SEC mode using the Agilent 1260 Infinity II Bio-inert LC. Also, 50 μL of complement component C5 at 31.6 μM (5.96 mg/mL) was injected onto a Superdex 200 Increase 5/150 column (GE Healthcare, Amersham, UK) at a flow rate of 0.35 mL/min. The mobile phase comprised 20 mM Tris pH 7.35, 75 mM NaCl, and 3% glycerol. The column elute was directly streamed to the SAXS capillary cell, and throughout the 15-min run, 900 frames of 1 s exposure were collected. After data reduction and radial averaging, the program CHROMIXS (*Panjkovich and Svergun, 2018*) was employed. Around 100 statistically similar buffer frames were selected and used for background subtraction of the sample frames from the chromatographic peak. This results in the final I(s) vs s scattering profiles, where $s = 4\pi\sin\theta/\lambda$, $2\theta$ is the scattering angle, and $\lambda = 1.24$ Å. The scattering data in the momentum transfer range $0.05 < s < 0.32$ $nm^{-1}$ were collected with a PILATUS 6M pixel detector at a distance of 3.1 m from the sample.

ATSAS 2.8 (*Franke et al., 2017*) was employed for further data analysis and modelling. The program PRIMUS (*Konarev et al., 2003*) was used to perform Guinier analysis ($\ln I(s)$ versus $s^2$) from which the radius of gyration, $R_G$, was determined. Distance probability functions, p(r), were calculated using the inverse Fourier transformation method implemented in GNOM (*Svergun, 1992*) that provided the maximum particle dimension, $D_{max}$. The concentration-independent molecular weight estimate, $MW_{VC}$, is based on the volume of correlation (*Rambo and Tainer, 2013*). The values are reported in *Supplementary file 1,* Table 3.1.

Theoretical scattering profiles were computed from X-ray coordinates using Crysol (*Svergun et al., 1995*), and SREFLEX (*Panjkovich and Svergun, 2016*) was used to refine the

models. For this, the program partitions the structure into pseudo-domains and hierarchically employs NMA to find the domain rearrangements minimising the discrepancy $\chi^2$ between the SAXS curve computed from the refined model and the experimental data. Because of the absence of electron density for the C345c domain in the C5-K8 complex structure, we included a round of restrained rigid body refinement followed by NMA to obtain an improved fit.

On the same day, MALLS data were collected with a separate SEC run under the same experimental conditions (set-up, buffer, run parameters, etc.). For this, a Wyatt Technologies miniDAWN TREOS MALLS detector coupled to an OptiLab T-Rex differential refractometer for protein concentration determination (dn/dc was taken as 0.185) was used. The MALLS system was calibrated relative to the scattering from toluene. The MWMALLS distribution of species eluting from the SEC column were determined with the Wyatt ASTRA7 software package.

The experimental SAXS data and the models derived from them were deposited to the Small Angle Scattering Biological Data Bank (SASBDB accession number SASDJA6).

## Hydrogen/deuterium mass spectrometry

A 6 µM of C5 was incubated with 10 µM of peptide (K8, K92, or K57) to achieve complex during deuterium exchange conditions. Then, 4 µL of C5 or the C5-peptide complex were diluted into 57 µL of 10 mM phosphate in $H_2O$ (pH 7.0) or into 10 mM phosphate in $D_2O$ (pD 7.0) at 25°C. The deuterated samples were then incubated for 0.5, 2, 15, and 60 min at 25°C. After the reaction, all samples were quenched by mixing at 1:1 (v/v) with a quench buffer (4 M guanidine hydrochloride, 250 mM Tris (2-carboxyethyl) phosphine hydrochloride, 100 mM phosphate) at 1°C. The final mixed solution was pH 2.5. The mixture was then immediately injected into the nanoAcquity HDX module (Waters Corp., Milford, MA) for peptic digest using an enzymatic online digestion column (Waters Corp., Milford, MA) in 0.2% formic acid in water at 20°C and with a flow rate of 100 µL/min. All deuterated time points and undeuterated controls were carried out in triplicate with blanks run between each data point.

Peptide fragments were then trapped using an Acquity BEH C18 1.7 µM VANGUARD chilled precolumn for 3 min. Peptides were eluted into a chilled Acquity UPLC BEH C18 1.7 µm 1.0 × 100 using the following gradient: 0 min, 5% B; 6 min, 35% B; 7 min, 40% B; 8 min, 95% B, 11 min, 5% B; 12 min, 95% B; 13 min, 5% B; 14 min, 9 5% B; and 15 min, 5% B (A: 0.2% HCOOH in H2O; B: 0.2% HCOOH) in acetonitrile. The trap and UPLC columns were both maintained at 0°C. Peptide fragments were ionised by positive electrospray into a Synapt G2-Si mass spectrometer (Waters Corp., Milford, MA). Data acquisition was run in ToF-only mode over an m/z range of 50–2000 Th using an MSE method (low collision energy, 4 V; high collision energy: ramp from 18 V to 40 V). Glu-1-Fibrinopeptide B peptide was used for internal lock mass correction. To avoid significant peptide carry-over between runs, the on-line Enzymate pepsin column (Waters Corp., Milford, MA) was washed three times with pepsin wash (0.8% formic acid, 1.5 M Gu-HCl, 4% MeOH) and a blank run was performed between each sample run.

MSE data from undeuterated samples of C5 were used for sequence identification using the Waters Protein Lynx Global Server 2.5.1 (PLGS). Ion accounting files for the three control samples were combined into a peptide list imported into DynamX v3.0 software (Waters Corp., Milford, MA). The output peptides were subjected to further filtering in DynamX. Filtering parameters used were minimum and maximum peptide sequence length of 4 and 25, respectively, minimum intensity of 1000, minimum MS/MS products of 2, minimum products per amino acid of 0.2, and a maximum MH + error threshold of 10 ppm. DynamX was used to quantify the isotopic envelopes resulting from deuterium uptake for each peptide at each time point. Furthermore, all the spectra were examined and checked visually to ensure correct assignment of m/z peaks and only peptides with a high signal to noise ratios were used for HDX-MS analysis.

Following manual filtration in DynamX, confidence intervals for differential HDX-MS (ΔHDX) measurements of individual time point were calculated using Deuteros (*Lau et al., 2019*) software. Only peptides which satisfied a ΔHDX confidence interval of 98% were considered significant. The ΔHDX was then plotted onto the C5 structure in Pymol.

## Surface plasmon resonance, single-cycle kinetics

On a Biacore 8K (GE Healthcare, Amersham, UK), human C5b was immobilised on a CM5 chip (GE Healthcare, Amersham, UK). Flow cells were activated using a standard immobilisation protocol: EDC/NHS was mixed at 1:1 ratio (flow rate, 10 µL/min; contact time, 30 s). C5b, at 1 µg/mL, or C5b-6 (Complement Technologies, Tyler, TX), at 2 µg/mL, in pH 4.5 sodium acetate buffer, were immobilised on flow cell two only (flow rate, 10 µL/min; contact time, 420 s). Finally, ethanolamine was applied to both flow cells (flow rate, 10 µL/min; contact time, 420 s). A final immobilisation level of 500–700 RUs was obtained for C5b and 1000–1150 RUs were obtained for C5b-6. Single-cycle kinetics were measured using a seven-point, threefold serial dilution (spanning a range of 1 µM to 1.4 nM) in HBS-EP buffer (GE Healthcare, Amersham, UK). A high flow rate of 40 µL/min was used, with a contact time of 300 s and a dissociation time of 2700 s. Binding to the reference surface was subtracted, and the data were fitted to a single-site binding model using Biacore evaluation software. All sensorgrams were inspected for evidence of mass transport limitation using the flow rate-independent component of the mass transfer constant (tc).

## Surface plasmon resonance, cross blocking

On a Biacore 8K (GE Healthcare, Amersham, UK), human C5 was amine coupled to a CM5 chip using the same protocol as for C5b. A final immobilisation level of approximately 1000–2000 RUs was obtained. For cross blocking, the surface was saturated with two sequential injections of a 20 µM knob domain solution in HBS-EP buffer (GE Healthcare, Amersham, UK) using a flow rate of 30 µL/min and contact time of 300 s. This was immediately followed with an injection of a second knob domain peptide, again at 20 µM in HBS-EP, with a flow rate of 30 µL/min and a contact time of 270 s, and the dissociation phase was measured for 600 s. Binding to the reference surface was subtracted, and sensorgrams were plotted in GraphPad Prism (GraphPad Software, San Diego, California USA, www.graphpad.com).

## Acknowledgements

We thank John Cashman for his help during crystallography screening. We are grateful to the three reviewers and editor for their valuable comments that helped to improve our manuscript.

## Additional information

### Competing interests

Charlotte M Deane: Reviewing editor, *eLife*. Alex Macpherson, Zainab Ahdash, James R Birtley, Monika-Sarah ED Schulze, Ben Holmes, Vladas Oleinikovas, James Snowden, Victoria Ellis, Alastair DG Lawson: employee of UCB and may hold shares and/or stock options. Tom Eirik Mollnes: T.E.M is a Board member of Ra Pharmaceuticals, Inc. The other authors declare that no competing interests exist.

### Funding

No external funding was received for this work.

### Author contributions

Alex Macpherson, Conceptualization, Data curation, Formal analysis, Validation, Investigation, Visualization, Methodology, Writing - original draft, Writing - review and editing; Maisem Laabei, Data curation, Formal analysis, Investigation; Zainab Ahdash, Formal analysis, Investigation, Methodology; Melissa A Graewert, Sarah A Robinson, Ben Holmes, Vladas Oleinikovas, Per H Nilsson, James Snowden, Victoria Ellis, Tom Eirik Mollnes, Dmitri Svergun, Investigation, Methodology; James R Birtley, Monika-Sarah ED Schulze, Susan Crennell, Formal analysis, Validation, Investigation, Methodology; Charlotte M Deane, Formal analysis, Investigation, Methodology, Writing - original draft; Alastair DG Lawson, Conceptualization, Resources, Formal analysis, Supervision, Validation, Investigation, Methodology, Writing - original draft, Project administration, Writing - review and editing; Jean MH van den Elsen, Conceptualization, Resources, Data curation, Formal analysis, Supervision, Funding

acquisition, Validation, Investigation, Visualization, Methodology, Writing - original draft, Project administration, Writing - review and editing

### Author ORCIDs
Alex Macpherson http://orcid.org/0000-0002-4508-5322
Maisem Laabei http://orcid.org/0000-0002-8425-3704
Zainab Ahdash http://orcid.org/0000-0002-4495-8689
James Snowden http://orcid.org/0000-0003-4855-7329
Jean MH van den Elsen https://orcid.org/0000-0002-0367-1956

### Decision letter and Author response
Decision letter https://doi.org/10.7554/eLife.63586.sa1
Author response https://doi.org/10.7554/eLife.63586.sa2

## Additional files

### Supplementary files
• Supplementary file 1. Section 1. Functional analysis. Table 1.1. Classical pathway C5b deposition ELISA; Table 1.2. Alternative pathway C5b deposition ELISA; Table 1.3. Inhibition of classical pathway-mediated C5a release; Table 1.4. Inhibition of alternative pathway-mediated C5a release; Table 1.5. Inhibition of classical pathway-mediated C9 deposition; Table 1.6. Inhibition of alternative pathway-mediated C9 deposition; Table 1.7. Inhibition of classical pathway haemolysis; Table 1.8. Inhibition of alternative pathway haemolysis. Section 2. Structural analysis. Table 2.1. Data collection and refinement statistics (molecular replacement); Table 2.2. Hydrogen bond interactions between K8 and C5; Table 2.3. Ionic interactions between K8 and C5; Table 2.4. Disulphide mapping of the K92 peptide; Table 2.5. Hydrogen bond interactions between K92 and C5; Table 2.6. Validation of molecular interactions by peptide mutagenesis analysis; Table 2.7. Individual, total, and average hydrogen bond persistence in a binding pose metadynamics simulation of the K8-C5 complex; Table 2.8. Individual, total, and average hydrogen bond persistence in a binding pose metadynamics simulation of the K92-C5 complex. Section 3. Solution structure analysis. Table 3.1. SAXS Summary data; Table 3.2. ΔHDX summary data. Section 4. Additional functional analyses. Table 4.1. SPR single-cycle kinetics of knob domains binding to human C5b; Table 4.2. SPR single-cycle kinetics of knob domains binding to human C5b-6.

• Transparent reporting form

### Data availability
Structural datasets presented in this study have been made publicly available in the Protein Data Bank (PDB) and Small Angle Scattering Biological Data Bank (SASBDB).

The following datasets were generated:

| Author(s) | Year | Dataset title | Dataset URL | Database and Identifier |
|---|---|---|---|---|
| Macpherson A, Elsen JM | 2021 | Crystal structure C5-K8 complex | https://www.rcsb.org/structure/7AD7 | RCSB Protein Data Bank, 7AD7 |
| Macpherson A, Elsen JM | 2021 | Crystal structure of C5-K92 complex | https://www.rcsb.org/structure/7AD6 | RCSB Protein Data Bank, 7AD6 |
| Macpherson A, Elsen JM, Graewert MA, Svergun D | 2020 | SAXS data and models of C5-bovine knob domain peptides | https://www.sasbdb.org/data/SASDJA6/ | Small Angle Scattering Biological Data Bank, SASDJA6 |

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
