## [Decision Letter]

Thank you for submitting your article "The allosteric modulation of Complement C5 by knob domain peptides" for consideration by *eLife*. Your article has been reviewed by three peer reviewers, and the evaluation has been overseen by John Kuriyan as the Reviewing Editor and the Senior Editor. The reviewers have opted to remain anonymous.

The reviewers have discussed the reviews with one another and the Reviewing Editor has drafted this decision to help you prepare a revised submission.

In this paper the authors present structural and biochemical evidence for how "knob" domain peptides, derived from cow CDR3s, bind to and inhibit complement protein C5. The authors use a range of biophysical techniques to rationalize the mechanism. This represents a new and versatile modality for complement inhibition. Further, the work identifies new epitopes for pharmacologic intervention in C5, and describes detailed biophysical properties of the CDR3 knob domains.

The overall response to your manuscript is favorable, but the review has raised a number of issues that you should addressed in a revised manuscript.

The underlying amount of data in the paper is substantial, but the data analysis and presentation are not optimal. The findings with respect to differential inhibition of the C5 convertases is interesting, but the mechanistic insight is limited and can be improved by comparison with prior work.

The reviewers have commented that the organization of the manuscript is confusing, and some crucial data, particularly structural details, are not effectively presented. The organization of the manuscript is confusing at multiple levels. The authors may consider presenting the SPR data earlier with the functional studies. Descriptions of structures are meandering. A number of sections seem out of place. Discussion should be better organized in terms of function, structure and mechanism. Most paragraphs do not contain a concluding sentence, leaving the reader wondering about the conclusion of the reasoning.

Please take this opportunity to revise the manuscript for overall clarity and coherence, in addition to responding to the specific points below.

Major concerns requiring additional experimental data:

1) Speculations on the importance of molecular interactions need to be experimentally supported, or need to be clearly indicate as speculations. In particular, mutagenesis studies are conspicuously absent. The two binding sites defined by crystallography are not validated by mutagenesis in the peptides although this appears straightforward with the methodology presented by authors. In particular, this could help to validate the observed K92 binding mode. Alternatively, a K92 structure would be very complementary, but perhaps experimentally difficult to achieve.

Major concerns that do not require new experimental data:

1) The interpretation of the SAXS data is rudimentary. The authors manage to fit the data in a reasonable manner by NMA for apo-C5 and C5-K92, but not for K8. It is suggested that flexibility of the C345c domain is the reason. Perhaps the fit to the data can be improved by switching to rigid body refinement, taking into account the known structural properties of C5 and considering the whole ensemble of known C5 conformations as starting model one at a time rather than relying on the NMA approach. If data cannot be fitted by a single model, ensemble optimization perhaps could model the system if C345c flexibility is the only problem. It may also be worth investigating whether the standard deviations for the K8-C5 SEC-SAXS data are correct. In other words, is the high chi^2^ value due to an inability to fit the curve or is it due to underestimated standard deviations?

2) Related to this, and also a serious concern due to the suggested importance of allosteric coupling between the binding sites for the knob peptides, it is surprising that no analysis of the overall conformation of C5-K8 and C5-K92, and comparison with known structures of C5 that clearly exhibit different overall conformations, is presented. Such an analysis would strengthen the manuscript substantially and possibly allow a better interpretation of the SPR and HDX data presented.

3) The crystal structures presented in this manuscript represent among the first structures of knob peptides derived from bovine immunoglobulins. Unfortunately, the manuscript presents only cursory analysis of the structures. The manuscript can be substantially strengthened by including the following analyses:

i) The crystal structure of the K92-C5 complex must be better documented. The difference mFo-DFc map presented at 1σ in Figure S2.4 is not that convincing, and 1σ is really low for a difference map whereas the 3σ used for K8-C5 is normal. What is the Rfree increase if K92 is omitted, and what is the average B-factor of K92 compared to that of C5? Is it possible that occupancy is significantly < 1? Perhaps it would be wise to down-weight the details concerning K92 binding and focus on the overall location and effects that can be detected by non-crystallographic methods.

ii) Structural comparisons with other cysteine-constrained peptides including natural toxins and engineered molecules, e.g, those from peptide libraries. It is unclear whether these inhibitors are structurally novel or similar to existing cysteine-constrained peptides. It is odd that the authors compared these peptides only with antibodies.

iii) Very few figures support the section describing the structures of the inhibitors and their interactions with C5. Minimally, the following figures should be included: comparisons of the epitopes with other inhibitors binding to C5; details of interactions between the peptides and C5.

iv) Figure 2 is not effective. Use stronger color contrasts. It is impossible to see 3-stranded β-sheet in panel c. Figure 3 is incomprehensible. Where are the paratopes?

4) The graphical presentation of the K8 and K92 interface can be much improved to show details of hydrogen bonds, electrostatic interactions and water mediated contacts, the latter must be well defined in the K8 complex. Furthermore, the hydrophobicity of the interface from the PISA analysis could be provided. Such improved figures would also improve the value of the rather detailed structure description provided in p 6-7, currently it is difficult to relate to without having the structures available.

5) Human C5b is not a well-behaved reagent and is not conformationally stable. It would be more relevant to conduct binding studies with the C5b6 complex, which is a standard reagent commercially available and stable in solution. As a minimum, the properties of C5b versus C5 should be discussed.

6) It is clear that the low resolution of the structure required modeling to be significantly aided by the disulfide mapping. The authors should comment on this caveat and its potential accuracy.

7) The manuscript should include a reaction scheme for C5 activation and indicate what the ELISA methods measure in such a figure. The section is accessible to only those familiar with C5, e.g. "C3b and C9 deposition", "no effect was observed in assays where the AP component was not isolated"

[Editors' note: further revisions were suggested prior to acceptance, as described below.]

Thank you for submitting your article "The allosteric modulation of Complement C5 by knob domain peptides" for consideration by *eLife*. Your article has been reviewed by three peer reviewers, and the evaluation has been overseen by a Reviewing Editor and John Kuriyan as the Senior Editor. The reviewers have opted to remain anonymous.

All three reviewers feel that the authors have done a good job in responding to the initial review, but have raised several points that should be addressed in a revised manuscript. No new data are necessary, and these comments can be addressed by textual changes.

Reviewer #1:

The authors aim at exploring the potential of four peptides derived from ultralong CDR3 regions in bovine antibodies binding complement C5 with high affinity as inhibitors of the terminal pathway of complement. Their efforts include an impressive combination of functional in vitro assays combined with biochemical/biophysical characterization including detailed structure determination of two C5-peptide complexes. This highly interdisciplinary approach represents a major strength of the study, also since the different kind of data all support the overall model put forward stating that allosteric effects underlie the ability of three of these so-called knob peptides to inhibit terminal pathway activity fully or partially.

One weakness of the study is that the structural analysis of inhibitor-C5 complex and comparison with all known structures involving C5 and C5b is incomplete. In a wider perspective, in the light of the clear differences with respect to inhibiting CP and AP C5 convertase observed for one of the peptides , the manuscript could also discuss in more details how their new results can help the community to understand better fundamental aspects such as: 1) C5 recognition by the two convertase; 2) The structural organization of the C5 convertases.

Development of complement therapeutics is a very active field and C5 inhibitors are at the core of these efforts. For this reason, the community may receive this study with quite some interest and the described peptides may become important reagents for functional studies of C5 convertase if they can be made generally available. The significance of work may be perceived even more positive, if a clear comparison with other well characterized C5 inhibitors was included. Furthermore, it could strengthen the general interest if the perspectives for in vivo use of these knob peptides were better discussed, especially if there are conditions where these peptides could have advantages over known C5 inhibitors

Reviewer #2:

Cow antibodies are unusual in having "ultralong" CDR3 regions which form independently folding "stalk" and "knob" minidomains which appear to be the antigen binding moiety. MacPherson et.al., for the first time, show structural data and functional activity of knob domain in the absence of the antibody. Because of the enormous diversity of disulfide bonding and loop patterns, immunization of cows present a novel mechanism to identify small, stable, target binding peptides. The authors present crystal structures of knob domains against complement factor C5 as well as several biochemical and biophysical assays evaluating their properties. They conclude that they have identified unique C5 epitopes, some of which function as allosteric modulators.

Reviewer #3:

Strengths

The authors employed an impressive array of functional and structural techniques to determine the structure and interactions of four knob domain peptides with C5. The crystal structures of two complexes revealed novel knob domain structures and their ability to form diverse interface topography. SAXS and HD exchange analyses complement and support the crystal structures. The knob domains seem to act as allosteric inhibitors and the structural perturbations by the knob domains strongly suggest long-range allosteric coupling in C5, which informs a new way to control C5. Taken together, these results establish the ability of the bovine immune system to generate knob domain peptides with diverse disulfide connectivities, 3D structure and binding interfaces as well as the utility of knob domain peptides as reagents to perturb protein functions and as potential therapeutics.

Weaknesses

The manuscript includes limited enzymology characterization of the inhibition (Figure 1). Effects of the inhibitors on Km and kcat of the reaction are undefined. Consequently, the authors concluded that peptides K8 and K92 were allosteric, non-competitive inhibitors based solely on their inability to fully inhibit C5 activation. They are vague about the mechanism of peptide K57 that shows full inhibition but still can be a non-competitive inhibitor. Indeed, based on binding competition between K57 and 92 (Figure 2) and the high similarity of HDX profiles (Figure 6), it is probable that the two peptides are both allosteric inhibitors, as the authors suggest under Discussion. Similarly, the authors did not fully utilize the structural information to define the inhibition mechanism, e.g. examining whether the inhibitors directly compete against C5 convertases based on the epitopes of the inhibitors and convertases.

The competitive binding experiments using SPR (Figure 2) were performed using 20 µM peptides, corresponding to 1,000 to >10,000 times higher concentration than their reported Kd values. The rationale is unclear. The use of such high concentrations raises the possibility that some of the observed reactions may be due to nonspecific binding, which makes it difficult to conclude whether there is allosteric inhibition among these peptides.

The manuscript includes unsupported interpretations of the roles of structural features to binding. "stabilized by an extensive network of 18 hydrogen bonds"; "important H-bonds"; the section entitled "The multi-purpose role of disulphide bonds". Many of their interpretations are speculative, not supported by experimental evidence.

---

## [Author Response]

The overall response to your manuscript is favorable, but the review has raised a number of issues that you should addressed in a revised manuscript.The underlying amount of data in the paper is substantial, but the data analysis and presentation are not optimal. The findings with respect to differential inhibition of the C5 convertases is interesting, but the mechanistic insight is limited and can be improved by comparison with prior work.The reviewers have commented that the organization of the manuscript is confusing, and some crucial data, particularly structural details, are not effectively presented. The organization of the manuscript is confusing at multiple levels. The authors may consider presenting the SPR data earlier with the functional studies.

These are valid points and we have now reorganised the manuscript. In particular, we have included more detailed figures highlighting the molecular interactions between the knob domain peptides and C5 in more detail (see new Figure 3 (previously Figure 4)). In addition, we have removed the confusing Figure 3 (now Figure 4) and included an overview of previously determined structures of C5-inhibitor complexes for better comparison of the K8 and K92 knob domain peptide binding modes. We have also taken on board your suggestion to present the SPR cross-blocking experiments at the beginning of the manuscript and moved them right behind the functional data (now Figure 2) in the revised manuscript.

Descriptions of structures are meandering. A number of sections seem out of place. Discussion should be better organized in terms of function, structure and mechanism. Most paragraphs do not contain a concluding sentence, leaving the reader wondering about the conclusion of the reasoning.

This is a good suggestion, and we have now reordered the discussion into headed sections for *structure*, *function* and *mechanism*, in order to improve the organisation of this section of manuscript and added concluding sentences.

Please take this opportunity to revise the manuscript for overall clarity and coherence, in addition to responding to the specific points below.Major concerns requiring additional experimental data:1) Speculations on the importance of molecular interactions need to be experimentally supported, or need to be clearly indicate as speculations. In particular, mutagenesis studies are conspicuously absent. The two binding sites defined by crystallography are not validated by mutagenesis in the peptides although this appears straightforward with the methodology presented by authors. In particular, this could help to validate the observed K92 binding mode. Alternatively, a K92 structure would be very complementary, but perhaps experimentally difficult to achieve.

This is a very important point raised the reviewers. We anticipated this need and have produced a number of K8 and K92 peptide mutants for a follow-up study, so have been able to include an additional section in the revised manuscript addressing the presentation of these data. As can be seen in the revised manuscript, we have now validated the binding modes observed in the crystal structures of the complexes between C5 and the K8 and K92 peptides. For K8 we assessed R23A and R32A mutants targeting the two salt-bridge interactions with C5. Whilst for R23A we observed only a modest 2-fold reduction in binding, R32A resulted in a >700-fold drop in K_D_. For K92 we targeted an H-bond interaction with C5 sustained by H25 and the neighbouring aromatic residues of the interface with W21A and F26A mutants. Whilst we were unable to produce the H25A mutant, the loss of aromatics in W21A and F26A markedly abridged affinity with an >1200-fold and >45-fold drop in K_D_, respectively. By including these data, we hope to have convinced the reviewers that we now have validated the presented knob domain peptide C5 complexes.

Major concerns that do not require new experimental data:1) The interpretation of the SAXS data is rudimentary. The authors manage to fit the data in a reasonable manner by NMA for apo-C5 and C5-K92, but not for K8. It is suggested that flexibility of the C345c domain is the reason. Perhaps the fit to the data can be improved by switching to rigid body refinement, taking into account the known structural properties of C5 and considering the whole ensemble of known C5 conformations as starting model one at a time rather than relying on the NMA approach. If data cannot be fitted by a single model, ensemble optimization perhaps could model the system if C345c flexibility is the only problem. It may also be worth investigating whether the standard deviations for the K8-C5 SEC-SAXS data are correct. In other words, is the high chi^2^ value due to an inability to fit the curve or is it due to underestimated standard deviations?

We thank the reviewer for this insightful suggestion. We included a round of restrained rigid body refinement followed by NMA and obtained a much-improved fit to the K8 data with Chi2=4.1 as indicated in the revised version of the manuscript. The resulting model gives therefore a good representation of K8 construct in solution. Given the absence of clear electron density for C345C in the crystal, we kept the caveat about the possible flexibility in the text.

2) Related to this, and also a serious concern due to the suggested importance of allosteric coupling between the binding sites for the knob peptides, it is surprising that no analysis of the overall conformation of C5-K8 and C5-K92, and comparison with known structures of C5 that clearly exhibit different overall conformations, is presented. Such an analysis would strengthen the manuscript substantially and possibly allow a better interpretation of the SPR and HDX data presented.

This is a useful suggestion and we have now replaced Figure 3 with a new figure (Figure 4 in the revised manuscript), presenting the K8 and K92 binding sites in comparison with the complexes of C5 with known modulators (OmCI, RaCI, SSL7, CVF, Cirp-T and SKY59, Figure 4A). In addition, we show a close-up of the overlapping binding sites of SSL7 and K92 in the same figure (Figure 4B) and present a potential mechanism by which K92 exerts its allosteric inhibitory effects via a ligand-induced conformational change (Figure 4C).

3) The crystal structures presented in this manuscript represent among the first structures of knob peptides derived from bovine immunoglobulins. Unfortunately, the manuscript presents only cursory analysis of the structures. The manuscript can be substantially strengthened by including the following analyses:i) The crystal structure of the K92-C5 complex must be better documented. The difference mFo-DFc map presented at 1σ in Figure S2.4 is not that convincing, and 1σ is really low for a difference map whereas the 3σ used for K8-C5 is normal. What is the Rfree increase if K92 is omitted, and what is the average B-factor of K92 compared to that of C5? Is it possible that occupancy is significantly < 1? Perhaps it would be wise to down-weight the details concerning K92 binding and focus on the overall location and effects that can be detected by non-crystallographic methods.

We have now included this analysis and show that whilst for K8 the Rfree increases from 23.36 to 27.08 with the removal of the peptide in the refinement, for K92 the Rfree only increases from 25.35 to 25.52. This is also reflected in the relative b-factor values of the peptide in the complex and we have added this information in the main text and also added a new supplementary figure. We trust that with the addition of the mutational data in the revised manuscript that the reviewers are satisfied with our conclusions.

ii) Structural comparisons with other cysteine-constrained peptides including natural toxins and engineered molecules, e.g, those from peptide libraries. It is unclear whether these inhibitors are structurally novel or similar to existing cysteine-constrained peptides. It is odd that the authors compared these peptides only with antibodies.

A structural search using the DALI server did not identify homology with any structures in the PDB, including cysteine rich peptides. We have included a paragraph in the Results and a new supplementary figure in the revised manuscript comparing the number of interactions made by knob domains compared to cyclic peptides, as described in: Malde, Hill, Iyer and Fairlie, 2019. The reason why we compare these peptides with antibody binding sites is because they are part of the bovine antibody repertoire.

iii) Very few figures support the section describing the structures of the inhibitors and their interactions with C5. Minimally, the following figures should be included: comparisons of the epitopes with other inhibitors binding to C5; details of interactions between the peptides and C5.

This is a valid point, and we have amended our figures accordingly as described above, under general response and major concerns point 2. In the revised manuscript we now present the K8 and K92 binding sites in comparison with the complexes of C5 with known modulators (OmCI, RaCI, SSL7, CVF, Cirp-T and SKY59, Figure 4a). As described above, under major concerns point 2, we have now updated Figure 2 (Figure 3 in the revised manuscript) to show the molecular interactions involved in the K8 and K92 complexes with C5 in more detail.

iv) Figure 2 is not effective. Use stronger color contrasts. It is impossible to see 3-stranded β-sheet in panel c. Figure 3 is incomprehensible. Where are the paratopes?

To improve the visibility of the K8 and K92 peptide with respect to the C5 background, we have presented the peptide in molecular surface rendering. We have now removed Figure 3 and replaced it with a structural comparison of the K8- and K95-C5 complexes with complexes of C5 with known modulators.

4) The graphical presentation of the K8 and K92 interface can be much improved to show details of hydrogen bonds, electrostatic interactions and water mediated contacts, the latter must be well defined in the K8 complex. Furthermore, the hydrophobicity of the interface from the PISA analysis could be provided. Such improved figures would also improve the value of the rather detailed structure description provided in p 6-7, currently it is difficult to relate to without having the structures available.

As mentioned under point 2, we have now replaced Figure 2 with a new figure detailing the molecular interactions involved in the binding of K8 and K92 to C5 (see Figure 3 in the revised manuscript) to accompany the description of the interactions. For K8 we have now also included the water-mediated interactions involved at the binding interface.

5) Human C5b is not a well-behaved reagent and is not conformationally stable. It would be more relevant to conduct binding studies with the C5b6 complex, which is a standard reagent commercially available and stable in solution. As a minimum, the properties of C5b versus C5 should be discussed.

We acknowledge the concern regarding the stability of C5b and have therefore performed additional SPR experiments to measure the binding affinity of the knob domain peptides to the C5b-6 complex (*n=4*). These new data are shown in the text, and support the original observations with C5b, although the K8 peptide, which was not able to bind to C5b, did bind C5b-6.

6) It is clear that the low resolution of the structure required modeling to be significantly aided by the disulfide mapping. The authors should comment on this caveat and its potential accuracy.

We have now added a comment to reflect this, and in addition report a wider set of statistics, b-factors and Rfree with and without peptide, in addition to the stringent map which we had previously shown. As mentioned above, we have now added a section on validation of the structures by mutagenesis, confirming the molecular interactions observed in the crystal structures.

7) The manuscript should include a reaction scheme for C5 activation and indicate what the ELISA methods measure in such a figure. The section is accessible to only those familiar with C5, e.g. "C3b and C9 deposition", "no effect was observed in assays where the AP component was not isolated"

This is a good point, and we have now included a schematic of the methods used in Figure 1 of the revised manuscript.

[Editors' note: further revisions were suggested prior to acceptance, as described below.]

Reviewer #1:The authors aim at exploring the potential of four peptides derived from ultralong CDR3 regions in bovine antibodies binding complement C5 with high affinity as inhibitors of the terminal pathway of complement. Their efforts include an impressive combination of functional in vitro assays combined with biochemical/biophysical characterization including detailed structure determination of two C5-peptide complexes. This highly interdisciplinary approach represents a major strength of the study, also since the different kind of data all support the overall model put forward stating that allosteric effects underlie the ability of three of these so-called knob peptides to inhibit terminal pathway activity fully or partially.One weakness of the study is that the structural analysis of inhibitor-C5 complex and comparison with all known structures involving C5 and C5b is incomplete. In a wider perspective, in the light of the clear differences with respect to inhibiting CP and AP C5 convertase observed for one of the peptides , the manuscript could also discuss in more details how their new results can help the community to understand better fundamental aspects such as: 1) C5 recognition by the two convertase; 2) The structural organization of the C5 convertases.

We are very pleased with the positive comments made by this reviewer and we agree with the point made about the comparison with all known C5 and C5b structures and their complexes. We now expand on how K8 and K57 might influence C5 recognition by the convertase, based on the CVF-C5 model and added a figure (Figure 4—figure supplement 1).

Development of complement therapeutics is a very active field and C5 inhibitors are at the core of these efforts. For this reason, the community may receive this study with quite some interest and the described peptides may become important reagents for functional studies of C5 convertase if they can be made generally available. The significance of work may be perceived even more positive, if a clear comparison with other well characterized C5 inhibitors was included. Furthermore, it could strengthen the general interest if the perspectives for in vivo use of these knob peptides were better discussed, especially if there are conditions where these peptides could have advantages over known C5 inhibitors

We fully agree with this reviewer and we have now emphasised these important aspects in the revised manuscript (throughout and in the conclusion).

Reviewer #3:StrengthsThe authors employed an impressive array of functional and structural techniques to determine the structure and interactions of four knob domain peptides with C5. The crystal structures of two complexes revealed novel knob domain structures and their ability to form diverse interface topography. SAXS and HD exchange analyses complement and support the crystal structures. The knob domains seem to act as allosteric inhibitors and the structural perturbations by the knob domains strongly suggest long-range allosteric coupling in C5, which informs a new way to control C5. Taken together, these results establish the ability of the bovine immune system to generate knob domain peptides with diverse disulfide connectivities, 3D structure and binding interfaces as well as the utility of knob domain peptides as reagents to perturb protein functions and as potential therapeutics.WeaknessesThe manuscript includes limited enzymology characterization of the inhibition (Figure 1). Effects of the inhibitors on Km and kcat of the reaction are undefined. Consequently, the authors concluded that peptides K8 and K92 were allosteric, non-competitive inhibitors based solely on their inability to fully inhibit C5 activation. They are vague about the mechanism of peptide K57 that shows full inhibition but still can be a non-competitive inhibitor. Indeed, based on binding competition between K57 and 92 (Figure 2) and the high similarity of HDX profiles (Figure 6), it is probable that the two peptides are both allosteric inhibitors, as the authors suggest under Discussion. Similarly, the authors did not fully utilize the structural information to define the inhibition mechanism, e.g. examining whether the inhibitors directly compete against C5 convertases based on the epitopes of the inhibitors and convertases.

We agree that some aspects of the inhibitory mechanisms displayed by the peptides prompts further investigation and this is the focus of our ongoing research. In the revised manuscript we now expand on the comparison of our knob domain peptides with other C5 inhibitors and their potential effect on the C5 convertases and also included a new figure (Figure 4—figure supplement 1) to this respect.

The competitive binding experiments using SPR (Figure 2) were performed using 20 µM peptides, corresponding to 1,000 to >10,000 times higher concentration than their reported Kd values. The rationale is unclear. The use of such high concentrations raises the possibility that some of the observed reactions may be due to nonspecific binding, which makes it difficult to conclude whether there is allosteric inhibition among these peptides.

Our crystal structures show that the K8 epitope is different to that of K92 and the competitive binding experiments show that the two peptides compete, which would indicate allostery. The 20 µM injections of peptide were used to rapidly saturate the surface, we performed two saturating injections of the first peptide to show that after saturation the mass does not increase with a second injection, which would indicate a non-specific component to the binding.

The manuscript includes unsupported interpretations of the roles of structural features to binding. "stabilized by an extensive network of 18 hydrogen bonds"; "important H-bonds"; the section entitled "The multi-purpose role of disulphide bonds". Many of their interpretations are speculative, not supported by experimental evidence.

We agree with the reviewer’s comments and have now removed these unquantified adjectives when describing interactions.